# DIANA with Compression for Distributed Variational Inequalities: Eliminating the Need to Transmit Full Gradients

## Abstract

Variational inequalities (VIs) are attracting increasing interest among machine learning (ML) researchers due to their applicability in numerous areas, such as empirical risk minimization (ERM) problems, adversarial learning, generative adversarial networks (GANs), and robust optimization. The growing volume of training data necessitates the use of advanced architectures beyond single-node computations. Distributed optimization has emerged as the most natural and efficient paradigm, enabling multiple devices to perform training simultaneously. However, this setup introduces a significant challenge: devices must exchange information with each other, which can substantially reduce the speed of learning. A standard approach to mitigating this issue involves the use of heuristics that allow only partial information transmission. State-of-the-art methods with compression for distributed VIs rely on variance reduction techniques, which makes them inapplicable to practical tasks due to full gradient computation and transmission. In this paper, we obviate the need to consider full gradient computations and introduce a novel algorithm for solving distributed variational inequalities. It combines the classical DIANA algorithm with the Extragradient technique. Additionally, we incorporate an error compensation mechanism, enabling our algorithm to handle the class of contractive compression operators, which are more practical for real-world applications. We provide a comprehensive theoretical analysis with near-optimal convergence guarantees and additionally outperform competitors in CNN and GAN training experiments.

## 1 Introduction

The variational inequalities (VIs) problem first attracted the attention of researchers in economics and game theory (Von Neumann & Morgenstern, 1953). Nowadays such a problem, due to its flexible nature, is used in a variety of other fields. In this paper, we consider it in the following form:

$$\text{find } z^* \in Z \text{ such that } \langle F(z^*), z - z^* \rangle \geqslant 0 \text{ for all } z \in Z, \tag{1}$$

where $F \colon \mathbb{R}^d \to \mathbb{R}$ is a convex and monotone operator. The general setting (1) can be specified to address particular types of problems, such as classical minimization, saddle point, and fixed point problems (Kinderlehrer & Stampacchia, 2000; Facchinei & Pang, 2003). We provide several examples to show aforementioned relations.

**Example 1.1.** Consider the following convex unconstrained minimization problem:

$$\min_{z \in \mathbb{R}^d} \left[ f(z) \right], \tag{2}$$

where $f$ represents the smooth convex objective function. We define $F(z) = \nabla f(z)$. Then $z^* \in \mathbb{R}^d$ is the solution of equation 1 if and only if $z^* \in \mathbb{R}^d$ is the solution of equation 2. In this way, the problem (2) can be considered as a variational inequality.

**Example 1.2.** Now consider the convex-concave saddle point problem:

$$\min_{x \in \mathbb{R}^{d_x}} \max_{y \in \mathbb{R}^{d_y}} \left[ f(x, y) \right], \tag{3}$$

where $f$ represents the convex-concave smooth objective function. In this setting, we define $F(z) = F(x, y) = [\nabla_x f(x, y), -\nabla_y f(x, y)]$. Then $z^* \in \mathbb{R}^{d_x} \times \mathbb{R}^{d_y}$ is the solution of equation 1 if and only

if $z^* \in \mathbb{R}^{d_x} \times \mathbb{R}^{d_y}$ is the solution of equation 3. Consequently, the problem (3) admits a reformulation as a variational inequality.

Highlighting the numerous applications of variational inequalities, their expressive power is evident in areas such as discriminative clustering (Xu et al., 2004), matrix factorization (Bach et al., 2008), image denoising (Esser et al., 2010; Chambolle & Pock, 2011), and robust optimization (Ben-Tal et al., 2009). Furthermore, the problem (1) has found applications in cutting-edge machine learning approaches, including adversarial training (Madry et al., 2017), reinforcement learning (Omidshafiei et al., 2017; Jin & Sidford, 2020), and generative adversarial networks (GANs) (Goodfellow et al., 2014). The study of classical techniques for variational inequalities, such as averaging and extrapolation, provides valuable tools for a deeper understanding of DL training (Daskalakis et al., 2017; Gidel et al., 2018; Liang & Stokes, 2019; Peng et al., 2020). In summary, variational inequalities have recently become a pivotal tool for the machine learning community. The development of methods for solving these problems is essential for achieving advanced results in various areas of ML in the future.

On the other hand, the increasing volume of data in training samples presents new challenges for the ML community. Indeed, relying on a single node for computation makes it impossible to train state-of-the-art models with the desired quality within a reasonable time frame. The distributed learning paradigm has emerged as the natural solution to this problem (Verbraeken et al., 2020; Konečný et al., 2016; McMahan et al., 2017). Formally, we can describe this setting by choosing $F$ in equation 1 of the following form:

$$F(z) = \frac{1}{M} \sum_{i=1}^{M} F_i(z), \tag{4}$$

where $M$ is the number of devices in the network, and $F_i(\cdot)$ is a local operator based on the data stored at the $i$-th device. Consider an empirical risk minimization problem

$$\min_{w \in \mathbb{R}^d} \frac{1}{M} \sum_{i=1}^{M} \ell_i(g(w, x_i), y_i) + \frac{\lambda}{2} \|w\|^2 \tag{5}$$

(with $g(w, x_i)$ as the output of the machine learning model, $\ell(g(w, x_i), y_i)$ — loss function, $w \in \mathbb{R}^d$ — the model parameters, $x_i$ and $y_i$ — input data and label, respectively). In Madry et al. (2017) it was reformulated as a more broader $\min \max$ problem

$$\min_w \max_\sigma \frac{1}{M} \sum_{i=1}^{M} \ell_i(g(w, x_i + \sigma), y_i) + \frac{\lambda_1}{2} \|w\|^2 - \frac{\lambda_2}{2} \|\sigma\|^2, \tag{6}$$

where $\sigma$ represents adversarial noise introduced to model data perturbations, and $\lambda_1, \lambda_2$ are regularization parameters. This formulation can be expressed as a variational inequality (see Example 1.2) with:

$$z = \begin{pmatrix} w \\ \sigma \end{pmatrix}, \quad F_i(z) = \begin{pmatrix} \nabla_w \ell_i(w, x_i + \sigma, y_i) + \lambda_1 w \\ -\nabla_\sigma \ell_i(w, x_i + \sigma, y_i) + \lambda_2 \sigma \end{pmatrix}.$$

However, despite this setting paralleling computations across $M$ nodes, there is no $M$-fold acceleration. This bottleneck arises from the need for devices to exchange information to solve a common problem. One successful approach to mitigate communication costs is to compress the transmitted data. The core idea behind this approach is that by utilizing a sufficiently large number of workers in the network, the error caused by partial information exchange decreases as the number of transmitted bits is reduced, resulting in faster training (Gorbunov et al., 2021a; Mishchenko et al., 2024).

Proceeding with the formal description of this approach, we introduce two general classes of compression operators.

**Definition 1.3** (Unbiased compression operator). We say that a (possibly) stochastic mapping $Q : \mathbb{R}^d \to \mathbb{R}^d$ is an unbiased compression operator if there exists a constant $\omega \geq 1$ such that for all $z \in \mathbb{R}^d$ it implies

$$\mathbb{E}Q(z) = z, \ \mathbb{E} \|Q(z)\|^2 \leq \omega \|z\|^2.$$

**Example 1.4.** A canonical example of a *unbiased* compression operator is the RAND$n$ mapping Karimireddy et al. (2019); Horváth et al. (2023), defined as follows: given a vector $x \in \mathbb{R}^d$, RAND$n$ randomly selects $n$ coordinates (uniformly without replacement) and returns their scaled identity projection:

$$\text{RAND}n(x) = \frac{d}{n} \cdot \text{Proj}_S(x),$$

where $S \subseteq [d]$ denotes the randomly selected subset of size $n$, and $\mathrm{Proj}_S(x)$ retains only the coordinates in $S$, setting the rest to zero.

**Definition 1.5** (Contractive compression operator). We say that a (possibly) stochastic mapping $Q : \mathbb{R}^d \to \mathbb{R}^d$ is a contractive compression operator if there exists a constant $0 < \alpha \leq 1$ such that for all $z \in \mathbb{R}^d$ it implies

$$\mathbb{E} \left\| Q(z) - z \right\|^2 \leq (1 - \alpha) \left\| z \right\|^2.$$

**Example 1.6.** An important example of a *biased* compression operator is the TOP$n$ compressor Stich et al. (2018); Karimireddy et al. (2019), which selects the $n$ coordinates of largest magnitude while discarding the remaining entries. Formally, it is defined as:

$$\mathrm{TOP}n(x) = \mathrm{Proj}_T(x),$$

where $T \subseteq [d]$ contains the indices corresponding to the $n$ largest absolute values in $x$. Unlike RAND$n$, this compressor introduces bias due to the deterministic selection of the most significant components.

Given the Examples 1.4, 1.6 of the compression operators, one can note that the RAND$n$ compressor satisfies Definition 1.3 with $\omega = \frac{d}{n}$, and TOP$n$ satisfies Definition 1.5 with $\alpha = \frac{n}{d}$ (Alistarh et al., 2018; Beznosikov et al., 2023).

In this work, we present a novel approach to solving distributed variational inequalities with compression, encompassing both unbiased and contractive classes of compressors.

## 1.1 BRIEF LITERATURE REVIEW

Starting with methods for solving variational inequalities, the standard choice is the SGDA (Fallah et al., 2020) and EXTRAGRADIENT (Korpelevich, 1976) methods, however the last one has more accurate theoretical confirmation. This approach has been extended into various stochastic (Juditsky et al., 2011; Mishchenko et al., 2020; Gorbunov et al., 2022; Medyakov et al., 2024) and distributed versions (Srivastava et al., 2011; Liu et al., 2020; Mukherjee & Chakraborty, 2020; Rogozin et al., 2021). In addition to EXTRAGRADIENT, the MIRROR-PROX method (Nemirovski, 2004) allows for the incorporation of non-Euclidean geometry. Other notable methods include FORWARD-BACKWARD-FORWARD (Tseng, 2000), DUAL EXTRAPOLATION (Nesterov, 2007), REFLECTED GRADIENT (Malitsky, 2015), and FORWARD-REFLECTED-BACKWARD (FORB) (Malitsky & Tam, 2020), each offering specific advancements.

Next, we review communication-efficient approaches that utilize unbiased compression operators (see Definition 1.3) for classical minimization problems (Example 1.1). A prominent example is QSGD (Alistarh et al., 2017), which introduced a variant of SGD (Robbins & Monro, 1951; Lan, 2020) with compressed updates. Other methods built upon SGD include (Bernstein et al., 2018; Khirirat et al., 2018; Konečnỳ & Richtárik, 2018). More advanced algorithms, such as DIANA (Mishchenko et al., 2024; Li & Richtárik, 2020), VR-DIANA (Horváth et al., 2023), FEDCOM-GATE (Haddadpour et al., 2021), and MARINA (Gorbunov et al., 2021a), provide estimates under stochastic distributed settings. DIANA is based on

Table 1: Comparison of the convergence results.

| | Algorithm | Setup | Convergence to optimum | No Full Grad.? |
|---|---|---|---|---|
| **Unbiased Compressors** | QSGD (Alistarh et al., 2017) | min | ✗ | ✓ |
| | DIANA (Mishchenko et al., 2020) | min | ✓ | ✓ |
| | MARINA (Gorbunov et al., 2021a) | min | ✓ | ✗ |
| | DASHA (Tyurin & Richtárik, 2022) | min | ✓ | ✓ |
| | MASHA1 (Beznosikov et al., 2022) | min max | ✓ | ✗ |
| | EG DIANA (Algorithm 1) | min max | ✓ | ✓ |
| **Contractive Compressors** | EF (Karimireddy et al., 2019) | min | ✗ | ✓ |
| | EF21 (Richtárik et al., 2021) | min | ✓ | ✓ |
| | MASHA2 (Beznosikov et al., 2022) | min max | ✓ | ✗ |
| | EG DIANA EF (Algorithm 2) | min max | ✓ | ✓ |

*Columns:* Setup = min if considered classical minimization problem, min max if considered VIs problem, Compression Boost = communication complexity of the algorithm and uncompressed training ratio, Contr. Comp. = whether the method works with contractive compressors, No Full Grad.? = whether the method requires full gradients transmission.

compressing the difference between two gradient approximations, while MARINA employs the variance reduction technique SARAH (Nguyen et al., 2017). The MARINA algorithm achieves optimal communication and oracle complexity for non-convex smooth objectives. However, it requires periodic exchanges of full gradients, which limits its practical application in federated settings. The DASHA method (Tyurin & Richtárik, 2022) addresses this limitation while maintaining optimal communication complexity.

Approaches that enable convergence with contractive compressors (see Definition 1.5) deserve special attention. The ERROR FEEDBACK technique was proposed in (Seide et al., 2014; Ström, 2015), with theoretical analysis provided in (Stich et al., 2018; Karimireddy et al., 2019). Later, (Richtárik et al., 2021) introduced an advanced variant of this technique, EF21. A comparison is presented in Table 1. Although compression techniques for minimization problems have been well-studied, their application to variational inequalities remains unexplored. Several works address the communication bottleneck using various approaches, such as local steps (Yuan et al., 2014; Hou et al., 2021; Deng & Mahdavi, 2021) and data similarity (Beznosikov et al., 2020; 2021). However, to the best of our knowledge, there are almost no works that utilize compression operators, with the exception of two papers.

In (Yuan et al., 2014), rounding to the nearest multiple of a certain quantity was proposed, but this compressor did not achieve convergence to the solution. The work (Beznosikov et al., 2022) introduced the MASHA1 algorithm, which achieves optimal communication complexity for unbiased compressors and is $(M\omega)^{1/2}$ times worse for contractive compressors. To handle contractive compressors, the authors of (Beznosikov et al., 2022) incorporated the ERROR FEEDBACK technique. However, MASHA2 is based on the variance reduction approach SVRG (Johnson & Zhang, 2013; Alacaoglu & Malitsky, 2022), which requires regular transmission of the full operator value to all devices. This is a significant limitation, as broadcasting such data can be extremely time-consuming and impractical in federated learning scenarios. Later, the authors of (Beznosikov & Gasnikov, 2022) extended MASHA1 to a similarity setup (Shamir et al., 2014; Arjevani & Shamir, 2015; Kovalev et al., 2022), and in (Beznosikov et al., 2024), they incorporated local steps (Khaled et al., 2020; Gorbunov et al., 2021b; Luo et al., 2025).

## 1.2 OUR CONTRIBUTIONS

• **Novel Approach.** In contrast to the MASHA algorithms, which require the full transmission of operators at each iteration, we propose a novel and more efficient approach that eliminates this necessity. Rather than employing the variance reduction mechanism of SVRG, we draw upon the principles of DIANA, a distributed optimization method designed to accommodate compressed communication. We introduce the EXTRAGRADIENT DIANA algorithm, specifically tailored to operate with unbiased compressors. Additionally, we incorporate the ERROR FEEDBACK mechanism, which ensures convergence even when contractive compressors are utilized. To the best of our knowledge, this work is the first to combine DIANA with ERROR FEEDBACK, not only in the context of extragradient methods but also in classical minimization settings, thereby opening new avenues for research in compressed gradient-based optimization.

• **Theoretical Analysis.** We provide a comprehensive theoretical analysis, considering a practically relevant setup with $M$ nodes in a distributed network. For the algorithm with ERROR FEEDBACK, we assume strongly monotone operators in (1) and $L_i$-Lipschitz operators in (4). For the algorithm without error compensation, we additionally assume $L$-Lipschitz continuity for whole operator in (4).

• **Experiments.** Our comprehensive experiments demonstrate great advantages of our proposed algorithm: it shows not only superior performance to MASHA2 for $\min\max$ problems, but also an improved convergence compared to EF21 and DIANA originally designed for standard minimization problems. We validate these claims through two benchmark studies. First, we reformulate a standard minimization task (image classification on CIFAR-10 using RESNET18) as a saddle-point problem, where our method achieves faster convergence than MASHA2 and higher accuracy. Moreover, it shows competitive results with DIANA. Second, we demonstrate practical scalability through STYLEGAN training on the I'M SOMETHING OF A PAINTER MYSELF dataset, where our algorithm significantly outperforms MASHA2 in terms of convergence and final metrics.

## 2 ALGORITHMS AND ANALYSIS

We begin with the list of assumptions regarding the operator in equation 1 and equation 4, under which we develop our theoretical analysis.

**Assumption 2.1.** Each operator $F_i$ is $L_i$-Lipschitz, i.e., it satisfies $\|F_i(z_1) - F_i(z_2)\| \leq L_i\|z_1 - z_2\|$ for any $z_1, z_2 \in Z$. We denote $\hat{L} = \sqrt{1/M \sum_{i=1}^{M} L_i^2}$.

**Assumption 2.2.** Full operator $F$ is $L$-Lipschitz, i.e., it satisfies $\|F(z_1) - F(z_2)\| \leq L\|z_1 - z_2\|$ for any $z_1, z_2 \in Z$.

As a practical example of a Lipschitz operator $F$, consider the gradient of an objective function. In the distributed setting, each agent may be associated with its own local gradient $F_i = \nabla f_i$, which is $L_i$-Lipschitz continuous (Richtárik & Takáč, 2016; Beznosikov et al., 2020; 2022).

**Assumption 2.3.** Full operator $F$ is $\mu$-strongly monotone, i.e., it satisfies $\langle F(z_1) - F(z_2), z_1 - z_2 \rangle \geqslant \mu \|z_1 - z_2\|^2$ for any $z_1, z_2 \in Z$.

This assumption is standard in the analysis of distributed optimization and variational inequality methods (Scutari et al., 2013; Koshal et al., 2016) for foundational uses of strong monotonicity in multi-agent and game-theoretic settings.

Now we move to the algorithm, which holds unbiased compression operators, i.e., compressors from Definition 1.3.

### 2.1 EXTRAGRADIENT DIANA FOR UNBIASED COMPRESSORS

In this section, we present the EXTRAGRADIENT DIANA algorithm (Algorithm 1).

Originally, the idea of DIANA (Mishchenko et al., 2024) originated from the problems of classical QSGD, where naive compression of the local gradient/operator is performed. Indeed, $F_i(z^*)$ is not equal to zero, then the variance of $Q(F_i(z^*))$ does not reduce in the neighborhood of the optimum, and the method does not converge (Beznosikov et al., 2023). Therefore, DIANA compresses not the operator itself, but its difference with a specially constructed memory sequence $h_i^k$ that is striving to $F_i(z^*)$. In this way, we apply a compression operator to the value that is going to zero, allowing the method to converge.

To better understand how this idea integrates with the EXTRAGRADIENT method, let us recall its original update steps:

$$z^{k+\frac{1}{2}} = z^k - \gamma F(z^k),$$
$$z^{k+1} = z^k - \gamma F(z^{k+\frac{1}{2}}).$$

In original DIANA, the step is performed using $g^k$, which serves as an unbiased estimator of the full operator (Line 14). We maintain this idea for the main step of the method (Line 15), and the server retains this estimation through the entire computing process. The question arises regarding what to employ for the step to the extrapolation point $z^{k+1/2}$. One can try to use a similar estimator of the full gradient at a different point, e.g. $z^k$, but this complicates the analysis. In our method, we propose a more natural solution, namely to complete this step using an extra vector $h_k$ (Line 4). In this case, the difference $(F_i(z^{k+1/2}) - h_i^k)$ appears in the proof and can be easily estimated due to its convergence to zero. These modifications allow us to leverage both the $h^k$ and $g^k$ sequences while preserving the core concept of variance reduction.

---

**Algorithm 1** EXTRAGRADIENT DIANA

1: **Input:** initial point $z^0 \in Z$, initial vectors $\{h_i^0\}_{i=1}^M$, each $h_i^0 \in Z$, $h^0 = \frac{1}{M}\sum_{i=1}^M h_i^0$, compressor $Q$ is under Definition 1.3, number or workers $M$, number of iterations $K$
2: **Parameters:** Stepsize $\gamma > 0$
3: **for** $k = 0, 1, 2, ..., K-1$ **do**
4: $\quad z^{k+\frac{1}{2}} = z^k - \gamma h^k$
5: $\quad$ The server broadcast $z^{k+\frac{1}{2}}$ to all workers
6: $\quad$ **for** $i = 1, 2, \ldots, M$ **in parallel do**
7: $\quad\quad$ Compute $F_i(z^{k+\frac{1}{2}})$
8: $\quad\quad \Delta_i^k = F_i(z^{k+\frac{1}{2}}) - h_i^k$
9: $\quad\quad$ Compress $\hat{\Delta}_i^k \sim Q(\Delta_i^k)$
10: $\quad\quad$ Grad correction $h_i^{k+1} = h_i^k + \frac{1}{1+\omega}\hat{\Delta}_i^k$
11: $\quad\quad$ Broadcast $\hat{\Delta}_i^k$ to the server
12: $\quad$ **end for**
13: $\quad \hat{\Delta}^k = \frac{1}{M}\sum_{i=1}^M \hat{\Delta}_i^k$
14: $\quad g^k = h^k + \hat{\Delta}^k$
15: $\quad z^{k+1} = z^k - \gamma g^k$
16: $\quad h^{k+1} = h^k + \frac{1}{1+\omega}\hat{\Delta}^k$
17: **end for**
18: **Output:** $z^K$

---

It is also important to highlight the communication aspects of our approach. Given a set of vectors $\Delta_1, \ldots, \Delta_M \in Z$, each device computes in parallel and transmits a compressed version $Q(\Delta_i)$ to the central server at each aggregation round $k$. Notably, we do not apply compression when the server sends the intermediate point $z^{k+1/2}$ back to the devices (Line 5), as our primary focus is on reducing the communication cost associated with the exchange of gradients (Kairouz et al., 2021).

We present the following definition to further quantify the communication efficiency of our approach.

**Definition 2.4.** Let us define the expected density of the transmission with an unbiased compressor as

$$q_\omega^{-1} = 1/bd \cdot \mathbb{E}\|Q(z)\|_{\text{bits}},$$

where $\|z\|_{\text{bits}}$ represents the number of bits required to encode the vector $z$, $b$ denotes the number of bits per floating-point value, and $d$ is the dimensionality of the problem (i.e., $bd = \|z\|_{\text{bits}}$). Similarly, we can define $q_\alpha$ for the contractive compressors.

Formally, we assume a synchronous setup with equivalent devices and communication channels. Transmitting one unit of data from devices to the server takes $\tau$ time units. If clients start communicating simultaneously, channel initialization is negligible. Thus, one round of communication takes $\tau N$, where $N$ is the amount of data sent per device. Consequently, we can infer that the communication complexity at each iteration scales as $\mathcal{O}\left(1/q_\omega\right)$ or $\mathcal{O}\left(1/q_\alpha\right)$. Note that for the aforementioned compressor operators TOP$n$ and RAND$n$, according to the introduced Definition 2.4, the following equality holds: $\omega = \frac{1}{\alpha} = q_\omega = q_\alpha = \frac{d}{n}$.

Having established the methodological framework, we now turn our attention to the theoretical analysis of our approach.

**Theorem 2.5.** *Suppose Assumptions 2.1, 2.2, 2.3 hold. Then for Algorithm 1 with*

$$\gamma \leq \min\left\{\frac{1}{4\omega\mu}, \frac{1}{8\sqrt{30}\omega L}, \frac{\sqrt{M}}{40\sqrt{6} \cdot \omega^{3/2}\hat{L}}\right\}$$

*and $\beta = \frac{1}{\omega}$ the following estimate is fulfilled:*

$$\mathbb{E}V^K \leq (1 - \gamma\mu)^K V^0,$$

*where*

$$V^k = \left\|z^k - z^*\right\|^2 + \frac{200\omega^2\gamma^2}{M}\frac{1}{M}\sum_{i=1}^{M}\left\|F_i(z^{k+\frac{1}{2}}) - h_i^k\right\|^2 + 40\omega\gamma^2\left\|F(z^{k+\frac{1}{2}}) - h^k\right\|.$$

**Corollary 2.6.** *Suppose the conditions of Theorem 2.5 hold. Then Algorithm 1 has*

$$\mathcal{O}\left(\left(\frac{\omega}{q_\omega} + \frac{\omega L}{\mu q_\omega} + \frac{\omega^{3/2}\hat{L}}{\mu q_\omega\sqrt{M}}\right)\log\left(\frac{1}{\varepsilon}\right)\right)$$

*communication complexity, where $q_\omega$ is the expected density and $\varepsilon$ represents the accuracy of the solution.*

*Remark* 2.7. One can note, that for practical compressors (Beznosikov et al., 2023) it implies $q_\omega \geq \omega$. In that way, our estimate can be written as

$$\widetilde{\mathcal{O}}\left(1 + \frac{L}{\mu} + \frac{\omega^{1/2}\hat{L}}{\mu\sqrt{M}}\right).$$

We obtained a convergence estimate without requiring full gradient transmission, significantly enhancing the applicability of compression methods for VIs. Although the estimate includes a non-optimal $\omega^{1/2}$ term in the numerator, we do not consider it critical, as the $\sqrt{M}$ term in the denominator dominates the convergence behavior.

Now we proceed with our second algorithm, which handles contractive compressors.

## 2.2 EXTRAGRADIENT DIANA FOR CONTRACTIVE COMPRESSORS

The analysis of distributed EXTRAGRADIENT with contractive compressors (see Definition 1.5) raises specific challenges. There arises a dot product of the following form $\langle g^k, z^{k+\frac{1}{2}} - z^*\rangle$, where $g^k$ is the full gradient estimator. In this way, we cannot address it straightforwardly using the variance reduction technique DIANA, due to the compressor's bias and non-fulfillment of $\mathbb{E}_Q\left[\left\langle g^k, z^{k+\frac{1}{2}} - z^*\right\rangle\right] = \left\langle F\left(z^{k+\frac{1}{2}}\right), z^{k+\frac{1}{2}} - z^*\right\rangle$. To adapt our method to this setting, one can apply the EF21 approach; however, estimating a dot product of the form $\langle F(z^{k+\frac{1}{2}}) - g^{k-1}, z^{k+\frac{1}{2}} - z^*\rangle$, complicates the analysis and does not offer the prospect of an optimal final convergence rate.

Therefore, we employ a more standard method in our approach, specifically the ERROR FEEDBACK scheme. In this section, we present the EXTRAGRADIENT DIANA WITH ERROR FEEDBACK method (Algorithm 2).

Let us focus on several details. We apply the classical ERROR FEEDBACK scheme, i.e., add the error compensation term to the compressed vector (Lines 9,10). It is important to note that such compensation is necessary only for the full gradient estimators $g^k$ that directly perform the main step of the method. For updating the auxiliary sequence $h^k$, this technique is redundant. In this way, we additionally compress $\left(F_i(z^{k+1/2}) - h_i^k\right)$ (Lines 11, 12). Such a modification leads to broadcasting both $\hat{\Delta}_i^k$ and $\widetilde{\Delta}_i^k$ from devices to the server at each iteration to update $g^k$ and $h^k$ (Lines 17, 19). Mention it is not critical, since the communication complexity at each iteration is still $\mathcal{O}\left(1/q\right)$.

Now we move to the theoretical convergence results of Algorithm 2.

**Theorem 2.8.** *Suppose Assumptions 2.1, 2.3 hold. Then for Algorithm 2 with*

$$\gamma \leq \min \left\{ \frac{\alpha}{8\mu}, \frac{\alpha^2}{16\sqrt{1772}\hat{L}} \right\}$$

*and $\beta = 1 - \frac{\alpha}{8}$ the following estimate is fulfilled:*

$$\mathbb{E}V^K \leq (1 - \gamma\mu)^K V^0,$$

*where*

$$
\begin{aligned}
V^k &= \left\| z^k - z^* \right\|^2 \\
&+ \frac{7088\gamma^2}{\alpha^3} \frac{1}{M} \sum_{i=1}^{M} \left\| F_i(z^{k+\frac{1}{2}}) - h_i^k \right\|^2 \\
&+ \frac{276\gamma^2}{\alpha} \frac{1}{M} \sum_{i=1}^{M} \left\| e_i^k \right\|.
\end{aligned}
$$

**Corollary 2.9.** *Suppose the conditions of Theorem 2.8 hold. Then Algorithm 2 has*

$$\mathcal{O}\left( \left( \frac{1}{q_\alpha \alpha} + \frac{\hat{L}}{\mu \alpha^2 q_\alpha} \right) \log\left( \frac{1}{\varepsilon} \right) \right)$$

*communication complexity, where $q_\alpha$ is the expected density and $\varepsilon$ represents accuracy of the solution.*

*Remark* 2.10. One can note, that for practical compressors (Beznosikov et al., 2023) it implies $q_\alpha \geq \frac{1}{\alpha}$. In that way, our estimate can be written as

$$\widetilde{\mathcal{O}}\left( 1 + \frac{\hat{L}}{\mu\alpha} \right).$$

**Algorithm 2** EXTRAGRADIENT DIANA WITH ERROR FEEDBACK

1: **Input:** initial point $z^0 \in Z$, initial vectors $\{h_i^0\}_{i=1}^M$, $\{e_i^0\}_{i=1}^M$, each $h_i^0, e_i^0 \in Z$, $h^0 = \frac{1}{M}\sum_{i=1}^M h_i^0$, compressor $Q$ is under Definition 1.5, number or workers $M$, number of iterations $K$
2: **Parameters:** Stepsize $\gamma > 0$, bias $0 < \beta < 1$
3: **for** $k = 0, 1, 2, ..., K-1$ **do**
4: $\quad z^{k+\frac{1}{2}} = z^k - \gamma h^k$
5: $\quad$ The server broadcast $z^{k+\frac{1}{2}}$ to all workers
6: $\quad$ **for** $i = 1, 2, \ldots, M$ **in parallel do**
7: $\quad\quad$ Compute $F_i(z^{k+\frac{1}{2}})$
8: $\quad\quad \Delta_i^k = F_i(z^{k+\frac{1}{2}}) - h_i^k$
9: $\quad\quad$ Compress $\hat{\Delta}_i^k \sim Q(\Delta_i^k + e_i^k)$
10: $\quad\quad e_i^{k+1} = e_i^k + \left( \Delta_i^k - \hat{\Delta}_i^k \right)$
11: $\quad\quad$ Compress $\widetilde{\Delta}_i^k \sim Q(\Delta_i^k)$
12: $\quad\quad$ Grad correction $h_i^{k+1} = h_i^k + \beta\widetilde{\Delta}_i^k$
13: $\quad\quad$ Broadcast $\hat{\Delta}_i^k, \widetilde{\Delta}_i^k$ to the server
14: $\quad$ **end for**
15: $\quad \hat{\Delta}^k = \frac{1}{M}\sum_{i=1}^M \hat{\Delta}_i^k$
16: $\quad \widetilde{\Delta}^k = \frac{1}{M}\sum_{i=1}^M \widetilde{\Delta}_i^k$
17: $\quad g^k = h^k + \hat{\Delta}^k$
18: $\quad z^{k+1} = z^k - \gamma g^k$
19: $\quad h^{k+1} = h^k + \beta\widetilde{\Delta}^k$
20: **end for**
21: **Output:** $z^K$

Comparing this rate with the result of Theorem 2.5 one can note that a deterioration occurs in $1/\alpha$ times. A similar effect appeared in the work (Beznosikov et al., 2022) when matching MASHA1 and MASHA2.

Providing a final discussion of the theoretical results, we note that our approach yields a worse estimate than the MASHA1 method (see Table 1). Comparing with OPTIMISTIC MASHA (Beznosikov & Gasnikov, 2022) and THREE PILLARS ALGORITHM (Beznosikov et al., 2024), we observe that these methods exploit the data similarity assumption. Without it, their estimates revert to those of the MASHA1 method. However, all these methods utilize a momentum parameter, as in the work (Alacaoglu & Malitsky, 2022) and require the transmission of the full gradients. In this regard, our method represents a significant improvement.

## 3 EXPERIMENTS

In this section, we present an empirical study of our methods, focusing on the image classification problem and generative network training. Our code is open-sourced[1].

### 3.1 CLASSIFICATION PROBLEM

**Setup.** The first part of the numerical experiments is conducted on the CIFAR-10 dataset Krizhevsky et al. (2009), which is widely used as a benchmark in the optimization community, consisting of 50,000 training and 10,000 test samples. Each sample is an $32 \times 32$ RGB image associated with one of ten class labels. The experiments are implemented in Python using the PyTorch library Paszke et al. (2017), leveraging both a single CPU (Intel Xeon 2.20 GHz) and a single GPU (NVIDIA Tesla P100) for computation. To emulate a distributed environment, we split batches

---

[1] https://anonymous.4open.science/r/EG-DIANA-exp

across multiple workers, simulating a decentralized optimization setting. The total runtime for all experiments is approximately 40 hours.

We address a classification task on the CIFAR-10 dataset using the RESNET18 architecture Meng et al. (2019), a standard benchmark model for evaluating optimization algorithms due to its balance of complexity and performance. To explore the robustness of the optimizers, we reformulate the standard minimization problem into a min-max optimization framework as described in equation 6. Experimental details are available in Appendix A.

### 3.2 RESULTS AND DISCUSSION

**Performance Comparison.** We compare four algorithms: EXTRAGRADIENT DIANA WITH ERROR FEEDBACK, MASHA2, EF21 and DIANA with TOP$n\%$ compressor. Figure 1 represents the convergence of methods at compression levels $1\%, 5\%, 10\%$, respectively. To facilitate comprehension, all graphs have been aligned according to the comparison and compression factors. We evaluate the convergence rate in terms of time and the total amount of information exchanged between workers, providing a pairwise comparison of the methods. The results are presented in the left and the right columns of Figure 1, demonstrating the superior convergence behavior regarding communication efficiency for our approach. We employ an unconventional time metric for convergence comparison to emphasize a critical aspect: the SOTA method MASHA2 requires full gradient computation and transmission, which effectively doubles the actual wall-clock time needed for model training compared to methods that rely on stochastic gradient estimates. This distinction is crucial for understanding the practical efficiency of the algorithms, as it highlights the computational overhead associated with MASHA2's approach.

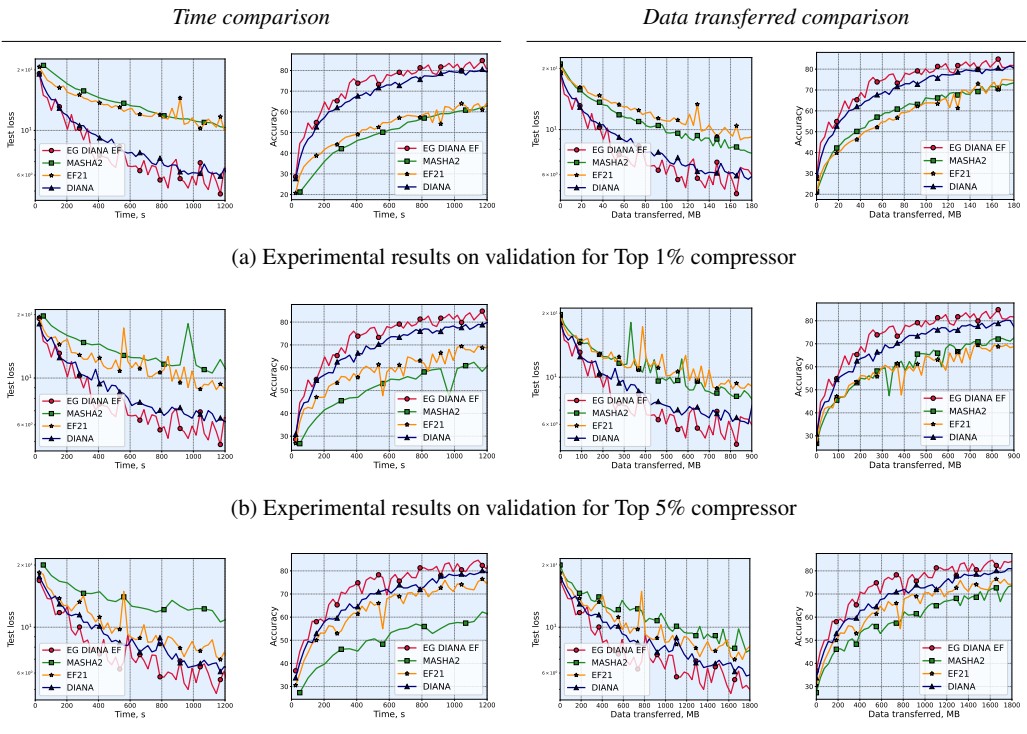

(a) Experimental results on validation for Top 1% compressor

(b) Experimental results on validation for Top 5% compressor

(c) Experimental results on validation for Top 10% compressor

Figure 1: Performance comparison of RESNET18 performance on CIFAR-10 with respect to the training time and amount of data transferred. Results for each type of compression value are presented in the rows.

**Discussion.** As anticipated by the theoretical analysis, MASHA2 demonstrates strong communication efficiency due to its use of full gradient information. However, in practice, Algorithm 2 achieves comparable and often superior performance in terms of both test accuracy and training speed, despite not relying on full gradient transmission. Moreover, our experimental results show that Algorithm 1 consistently outperforms distributed optimization methods such as EF21 and DIANA. This confirms the practical effectiveness of our approach.

## 3.3 GAN TRAINING

In this section, we present the experiments conducted using the GAN architecture, for which the loss functions are naturally $\min\max$ problems. Thus, our variational inequality approach serves as a common method to solve them. We implement STYLEGAN (introduced by Karras (2019)): a generative adversarial network that enables the generation of high-quality images while controlling the style of the generated images at various levels of detail.

**Setup.** Our experiments were conducted using the basic STYLEGAN implementation (. The model was trained on the I'M SOMETHING OF A PAINTER MYSELF dataset, which comprises 300 Monet paintings sized 256×256 and 7028 photos in the same format. The training process took place on the same hardware as in the previous section. We train the generator (specifically, the parameters of 2 generators for our architecture) with an extrapolation step (see Line 9) and subsequently make the «whole» step (see Line 18) for both discriminators. Note that, to maintain a variational inequality representation of the loss, an equal number of steps were taken for both generators and discriminators.

**Optimizers and Experimental Design.** Training a GAN is highly computationally expensive, thus we assumed in our experiments $M = 5$ number of clients and trained the model for only 160 epochs (however, this did not have a significant impact on the final results). Once again, we choose MASHA2 and EXTRAGRADIENT DIANA WITH ERROR FEEDBACK as optimizers, and TOP10% as a compressor. They demonstrated the best performance in terms of time in the previous section and exhibited nearly the same behavior regarding transmitted information. To maintain fairness, we set the following hyperparameters for both optimizers: i) learning rate: $5 \cdot 10^{-4}$; ii) exponential moving average (EMA) decay rate 0.99.

## 3.4 RESULTS AND DISCUSSION

Firstly, we present the generators and discriminators loss function graphs, as well as the image generation results for the best model in Figure 2. Secondly, we evaluate the models using the Fréchet Inception Distance (FID) metric (lower - better): i) **10.32** for EXTRAGRADIENT DIANA WITH ERROR FEEDBACK; ii) **24.47** for MASHA2.

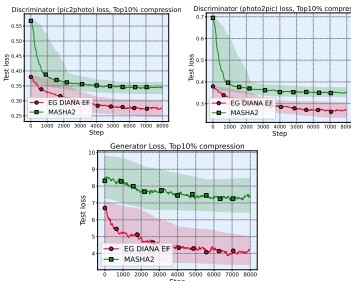
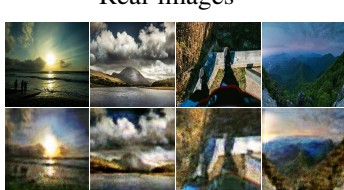

Real images

Generated images

Figure 2: Comparison of train losses for generator (averaged generators loss) and discriminators for STYLEGAN architecture under TOP10% compression (left half) and obtained results for generation on some examples pictures (right half).

**Performance comparison.** On Figure 2 we observe that Algorithm 2 exhibits more stable training dynamics compared to MASHA2. The generator and discriminator losses for DIANA converge more smoothly, indicating a balanced training process. In contrast, MASHA2 fluctuates around a local optimum due to its full gradient updates (see generator graph). Moreover, it can be observed from the FID metric evaluation that the STYLEGAN with full gradient updates does not achieve the desired quality of restored images.

**Discussion.** Despite the fact that I'M SOMETHING OF A PAINTER MYSELF... is 7 times smaller than CIFAR-10 dataset in terms of the number of classes, it comprises images of significantly higher resolution (by a factor of 16), making it an excellent benchmark for evaluating optimizer performance on more complex, high-dimensional data.
GAN architectures are well-known for their highly complex loss function landscapes, which makes them especially sensitive updates based on of full gradients. Although the error-feedback mechanism helps MASHA2 cope with compressed gradients, the variance reduction scheme inevitably drives the algorithm into local optima, making it inapplicable for real-world applications. In contrast, proposed Algorithm 2 eliminates the need for full gradient computation entirely. This not only leads to more stable training dynamics but also enables the achievement of superior metric values, demonstrating its effectiveness in handling the challenging optimization landscape inherent to GANs.

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

## A  EXPERIMENTAL DETAILS

The implementation of the stochastic versions of extragradient-based optimization algorithms discussed in this work is publicly available in an anonymized repository. The full source code can be accessed by link: `https://anonymous.4open.science/r/EG-DIANA-exp`.

### A.1  CLASSIFICATION PROBLEM

We compare two algorithms: EXTRAGRADIENT DIANA WITH ERROR FEEDBACK and MASHA WITH ERROR FEEDBACK. Both algorithms are evaluated in terms of information transferred (communication efficiency) and training time. The experiments are conducted with the following setup:

- number of workers ($M$): 10;
- learning rate: 0.01 for both optimizers;
- regularization parameters $\lambda_1 = \lambda_2 = 0.0005$.

**Compressor.**  Due to different sizes of inputs, we implement TOP$k\%$, which retains the largest $k\%$ values (in absolute) of a tensor. For MASHA WITH ERROR FEEDBACK, the server compressor is set to the identity one to ensure a fair comparison with EXTRAGRADIENT DIANA WITH ERROR FEEDBACK, which does not employ server compression.

**Hyperparameter Tuning**  To ensure a fair comparison, we conducted a grid search for key hyperparameters:

- for EXTRAGRADIENT DIANA WITH ERROR FEEDBACK, we tuned $\beta$ over the range $[0, 1]$;
- for MASHA WITH ERROR FEEDBACK, we tuned $\tau$ over a similar range.

The optimal values were found to be $\beta = 0.5$ for EXTRAGRADIENT DIANA WITH ERROR FEEDBACK and $\tau = 0.1$ for MASHA WITH ERROR FEEDBACK. These values were selected based on methods ability to achieve the best convergence and stability during training.

## B  NOTATION AND GENERAL INEQUALITIES

• **Notation.**  There we present notation used in the work: $\mathbb{E}[\xi]$ denotes the expected value of a random vector $\xi \in \mathbb{R}^d$, $\mathbb{E}_\eta[\xi]$ refers to the expected value of a random vector $\xi \in \mathbb{R}^d$ conditioned on a random vector $\eta \in \mathbb{R}^d$, $\|x\| = \|x\|_2 = \sqrt{\langle x, x \rangle}$ represents the Euclidean norm of the vector $x \in \mathbb{R}^d$.

• **General inequalities.**  We introduce important inequalities that are used in further proofs. Let $a_1, a_2 \in \mathbb{R}^d$, then

$$\|a_1 + a_2\|^2 \leqslant (1+s)\|a_1\|^2 + (1+s^{-1})\|a_2\|^2, \qquad \text{(Jensen)}$$

$$2\langle a_1, b_1 \rangle \leqslant \|a_1\|^2 \|b_1\|^2. \qquad \text{(CB)}$$

## C  PROOF OF THEOREM 2.5

**Theorem 2.5**  *Suppose Assumptions 2.1, 2.2, 2.3 hold.  Then for Algorithm 1 with* $\gamma \leq$

$$\min\left\{ \frac{1}{4\omega\mu}, \frac{1}{8\sqrt{30}\omega L}, \frac{\sqrt{M}}{40\sqrt{6}\cdot\omega^{3/2}\sqrt{\frac{1}{M}\sum\limits_{i=1}^{M} L_i^2}} \right\} \text{ and } \beta = \frac{1}{\omega} \text{ the following estimate is fulfilled:}$$

$$\mathbb{E}V^K \leq (1 - \gamma\mu)^K V^0,$$

*where* $V^k = \left\|z^k - z^*\right\|^2 + \frac{200\omega^2\gamma^2}{M}\frac{1}{M}\sum\limits_{i=1}^{M}\left\|F_i(z^{k+\frac{1}{2}}) - h_i^k\right\|^2 + 40\omega\gamma^2\left\|F(z^{k+\frac{1}{2}}) - h^k\right\|.$

*Proof.*  We start with the classicalal for EXTRAGRADIENT proofs equation for arbitrary $z, v$ and $u$:

$$\|z - u\|^2 = \|v - u\|^2 + 2\langle z - v, z - u \rangle - \|z - v\|^2.$$

For $z = z^{k+\frac{1}{2}}$, $u = z^*$ and $v = z^k$:

$$\left\|z^{k+\frac{1}{2}} - z^*\right\|^2 = \left\|z^k - z^*\right\|^2 + 2\left\langle z^{k+\frac{1}{2}} - z^k, z^{k+\frac{1}{2}} - z^* \right\rangle - \left\|z^{k+\frac{1}{2}} - z^k\right\|^2.$$

Using the same for $z = z^{k+1}$ with $u = z^*$ and $v = z^{k+\frac{1}{2}}$,

$$\left\| z^{k+1} - z^* \right\|^2 = \left\| z^{k+\frac{1}{2}} - z^* \right\|^2 + 2 \left\langle z^{k+1} - z^{k+\frac{1}{2}}, z^{k+\frac{1}{2}} - z^* \right\rangle + \left\| z^{k+\frac{1}{2}} - z^{k+1} \right\|^2.$$

After the summation with the previous equality, we obtain

$$
\begin{aligned}
\left\| z^{k+1} - z^* \right\|^2 &= \left\| z^k - z^* \right\|^2 + 2 \left\langle z^{k+1} - z^k, z^{k+\frac{1}{2}} - z^* \right\rangle \\
&\quad + \left\| z^{k+\frac{1}{2}} - z^{k+1} \right\|^2 - \left\| z^{k+\frac{1}{2}} - z^k \right\|^2.
\end{aligned}
\tag{7}
$$

Let's work with the term $\left\| z^{k+1} - z^{k+\frac{1}{2}} \right\|^2$:

$$
\begin{aligned}
\left\| z^{k+1} - z^{k+\frac{1}{2}} \right\|^2 &= \gamma^2 \left\| g^k - h^k \right\|^2 \\
&= \gamma^2 \left\| \frac{1}{M} \sum_{i=1}^M Q\left( F_i(z^{k+\frac{1}{2}}) - h_i^k \right) \right\|^2 \\
&= \gamma^2 \frac{1}{M^2} \sum_{i=1}^M \left\| Q\left( F_i(z^{k+\frac{1}{2}}) - h_i^k \right) \right\|^2 \\
&\quad + \gamma^2 \frac{1}{M^2} \sum_{i \neq j}^M \left\langle Q\left( F_i(z^{k+\frac{1}{2}}) - h_i^k \right), Q\left( F_j(z^{k+\frac{1}{2}}) - h_j^k \right) \right\rangle \\
&= \gamma^2 \frac{\omega}{M^2} \sum_{i=1}^M \left\| F_i(z^{k+\frac{1}{2}}) - h_i^k \right\|^2 \\
&\quad + \gamma^2 \frac{1}{M^2} \sum_{i \neq j}^M \left\langle Q\left( F_i(z^{k+\frac{1}{2}}) - h_i^k \right), Q\left( F_j(z^{k+\frac{1}{2}}) - h_j^k \right) \right\rangle.
\end{aligned}
\tag{8}
$$

Due to unbiasdness of the operator $Q$ we can take the expectation with respect to $Q$ and get

$$
\begin{aligned}
\mathbb{E}_Q &\left[ \sum_{i \neq j}^M \left\langle Q\left( F_i(z^{k+\frac{1}{2}}) - h_i^k \right), Q\left( F_j(z^{k+\frac{1}{2}}) - h_j^k \right) \right\rangle \right] \\
&= \sum_{i \neq j}^M \left\langle F_i(z^{k+\frac{1}{2}}) - h_i^k, F_j(z^{k+\frac{1}{2}}) - h_j^k \right\rangle \\
&= \left\| \sum_{i=1}^M \left( F_i(z^{k+\frac{1}{2}}) - h_i^k \right) \right\|^2 - \sum_{i=1}^M \left\| F_i(z^{k+\frac{1}{2}}) - h_i^k \right\|^2 \\
&= M^2 \left\| F(z^{k+\frac{1}{2}}) - h^k \right\|^2 - \sum_{i=1}^M \left\| F_i(z^{k+\frac{1}{2}}) - h_i^k \right\|^2.
\end{aligned}
\tag{9}
$$

Now we insert this result into equation 8 and simplify:

$$
\begin{aligned}
\mathbb{E} \left\| z^{k+\frac{1}{2}} - z^{k+1} \right\|^2 &\leq \gamma^2 \frac{\omega}{M^2} \sum_{i=1}^M \mathbb{E} \left\| F_i(z^{k+\frac{1}{2}}) - h_i^k \right\|^2 \\
&\quad + \gamma^2 \mathbb{E} \left\| F(z^{k+\frac{1}{2}}) - h^k \right\|^2 - \frac{\gamma^2}{M^2} \sum_{i=1}^M \mathbb{E} \left\| F_i(z^{k+\frac{1}{2}}) - h_i^k \right\|^2 \\
&= \gamma^2 \frac{\omega - 1}{M^2} \sum_{i=1}^M \mathbb{E} \left\| F_i(z^{k+\frac{1}{2}}) - h_i^k \right\|^2 + \gamma^2 \mathbb{E} \left\| F(z^{k+\frac{1}{2}}) - h^k \right\|^2.
\end{aligned}
\tag{10}
$$

Now we work with dot product term in equation 7 and take the expectation with respect to $Q$:

$$
\begin{aligned}
2\mathbb{E}_Q &\left[ \left\langle z^{k+1} - z^k, z^{k+\frac{1}{2}} - z^* \right\rangle \right] \\
&= -2\gamma \mathbb{E}_Q \left[ \left\langle g^k, z^{k+\frac{1}{2}} - z^* \right\rangle \right]
\end{aligned}
$$

$$
\begin{aligned}
&= -2\gamma \mathbb{E}_Q \left[ \left\langle h^k + \frac{1}{M} \sum_{i=1}^{M} Q\left( F_i(z^{k+\frac{1}{2}}) - h_i^k \right), z^{k+\frac{1}{2}} - z^* \right\rangle \right] \\
&= -2\gamma \left\langle F(z^{k+\frac{1}{2}}), z^{k+\frac{1}{2}} - z^* \right\rangle \\
&= -2\gamma \left\langle F(z^{k+\frac{1}{2}}) - F(z^*), z^{k+\frac{1}{2}} - z^* \right\rangle - 2\gamma \underbrace{\left\langle F(z^*), z^{k+\frac{1}{2}} - z^* \right\rangle}_{\leq 0} \\
&\overset{\text{Ass. 2.3}}{\leq} -2\gamma\mu \left\| z^{k+\frac{1}{2}} - z^* \right\|^2 .
\end{aligned}
$$

Substituting this result with equation 10 into equation 7 we get

$$
\begin{aligned}
\mathbb{E}\left\| z^{k+1} - z^* \right\|^2 &= \mathbb{E}\left\| z^k - z^* \right\|^2 - 2\gamma\mu \mathbb{E}\left\| z^{k+\frac{1}{2}} - z^* \right\|^2 \\
&\quad + \gamma^2 \frac{\omega - 1}{M^2} \sum_{i=1}^{M} \mathbb{E}\left\| F_i(z^{k+\frac{1}{2}}) - h_i^k \right\|^2 + \gamma^2 \mathbb{E}\left\| F(z^{k+\frac{1}{2}}) - h^k \right\|^2 \\
&\quad - \mathbb{E}\left\| z^{k+\frac{1}{2}} - z^k \right\|^2 \\
&\overset{CB}{\leq} (1 - \gamma\mu)\mathbb{E}\left\| z^k - z^* \right\|^2 + \gamma^2 \frac{\omega - 1}{M^2} \sum_{i=1}^{M} \mathbb{E}\left\| F_i(z^{k+\frac{1}{2}}) - h_i^k \right\|^2 \\
&\quad + \gamma^2 \mathbb{E}\left\| F(z^{k+\frac{1}{2}}) - h^k \right\|^2 - (1 - \gamma\mu)\mathbb{E}\left\| z^{k+\frac{1}{2}} - z^k \right\|^2 . \quad (11)
\end{aligned}
$$

Firstly, we derive a recursion on the $\left\| F(z^{k+\frac{3}{2}}) - h^{k+1} \right\|^2$:

$$
\begin{aligned}
\left\| F(z^{k+\frac{3}{2}}) - h^{k+1} \right\|^2 &= \left\| F(z^{k+\frac{3}{2}}) - h^k - \beta \frac{1}{M} \sum_{i=1}^{M} Q\left( F_i(z^{k+\frac{1}{2}}) - h_i^k \right) \right\|^2 \\
&\overset{Jensen}{\leq} (1 + c_1^{-1}) \left\| F(z^{k+\frac{3}{2}}) - F(z^{k+\frac{1}{2}}) \right\|^2 \\
&\quad + (1 + c_1) \left\| F(z^{k+\frac{1}{2}}) - h^k - \beta \frac{1}{M} \sum_{i=1}^{M} Q\left( F_i(z^{k+\frac{1}{2}}) - h_i^k \right) \right\|^2 \\
&\overset{\text{Ass. 2.2}}{\leq} (1 + c_1^{-1}) L^2 \left\| z^{k+\frac{3}{2}} - z^{k+\frac{1}{2}} \right\|^2 \\
&\quad + (1 + c_1) \left\| F(z^{k+\frac{1}{2}}) - h^k \right\|^2 \\
&\quad + (1 + c_1)\beta^2 \left\| \frac{1}{M} \sum_{i=1}^{M} Q\left( F_i(z^{k+\frac{1}{2}}) - h_i^k \right) \right\|^2 \\
&\quad - 2(1 + c_1)\beta \left\langle F(z^{k+\frac{1}{2}}) - h^k, \frac{1}{M} \sum_{i=1}^{M} Q\left( F_i(z^{k+\frac{1}{2}}) - h_i^k \right) \right\rangle \\
&\overset{\substack{\text{Def. 1.3}\\(i)}}{\leq} (1 + c_1^{-1}) L^2 \left\| z^{k+\frac{3}{2}} - z^{k+\frac{1}{2}} \right\|^2 \\
&\quad + (1 + c_1)(1 - 2\beta) \left\| F(z^{k+\frac{1}{2}}) - h^k \right\|^2 \\
&\quad + (1 + c_1)\beta^2 \left\| \frac{1}{M} \sum_{i=1}^{M} Q\left( F_i(z^{k+\frac{1}{2}}) - h_i^k \right) \right\|^2 \\
&\overset{8,9}{\leq} (1 + c_1^{-1}) L^2 \left\| z^{k+\frac{3}{2}} - z^{k+\frac{1}{2}} \right\|^2 \\
&\quad + (1 + c_1)(1 - 2\beta) \left\| F(z^{k+\frac{1}{2}}) - h^k \right\|^2
\end{aligned}
$$

$$+(1+c_1)\beta^2\frac{\omega-1}{M^2}\sum_{i=1}^{M}\left\|F_i(z^{k+\frac{1}{2}})-h_i^k\right\|^2$$

$$+(1+c_1)\beta^2\left\|F(z^{k+\frac{1}{2}})-h^k\right\|^2$$

$$=\quad(1+c_1^{-1})L^2\left\|z^{k+\frac{3}{2}}-z^{k+\frac{1}{2}}\right\|^2$$

$$+(1+c_1)(1-\beta)^2\left\|F(z^{k+\frac{1}{2}})-h^k\right\|^2$$

$$+(1+c_1)\beta^2\frac{\omega-1}{M^2}\sum_{i=1}^{M}\left\|F_i(z^{k+\frac{1}{2}})-h_i^k\right\|^2,\qquad(12)$$

where in $(i)$ we take expectation $\mathbb{E}_Q$ with respect to $Q$. Secondly, we obtain a recursion on the $\frac{1}{M}\sum_{i=1}^{M}\left\|F_i(z^{k+\frac{3}{2}})-h_i^{k+1}\right\|^2$:

$$\frac{1}{M}\sum_{i=1}^{M}\left\|F_i(z^{k+\frac{3}{2}})-h_i^{k+1}\right\|^2\quad=\quad\frac{1}{M}\sum_{i=1}^{M}\left\|F_i(z^{k+\frac{3}{2}})-h_i^k-\beta Q\left(F_i(z^{k+\frac{1}{2}})-h_i^k\right)\right\|^2$$

$$\overset{Jensen}{\le}\quad(1+c_2^{-1})\frac{1}{M}\sum_{i=1}^{M}\left\|F_i(z^{k+\frac{3}{2}})-F_i(z^{k+\frac{1}{2}})\right\|^2$$

$$+(1+c_2)\frac{1}{M}\sum_{i=1}^{M}\left\|F_i(z^{k+\frac{1}{2}})-h_i^k\right.$$

$$\left.-\beta Q\left(F_i(z^{k+\frac{1}{2}})-h_i^k\right)\right\|^2$$

$$\overset{Ass.\ 2.1}{\le}\quad(1+c_2^{-1})\frac{1}{M}\sum_{i=1}^{M}L_i^2\left\|z^{k+\frac{3}{2}}-z^{k+\frac{1}{2}}\right\|^2$$

$$+(1+c_2)\frac{1}{M}\sum_{i=1}^{M}\left\|F_i(z^{k+\frac{1}{2}})-h_i^k\right\|^2$$

$$+(1+c_2)\beta^2\frac{1}{M}\sum_{i=1}^{M}\left\|Q\left(F_i(z^{k+\frac{1}{2}})-h_i^k\right)\right\|^2$$

$$-2(1+c_2)\beta$$

$$\cdot\frac{1}{M}\sum_{i=1}^{M}\left\langle F_i(z^{k+\frac{1}{2}})-h_i^k,Q(F_i(z^{k+\frac{1}{2}})-h_i^k)\right\rangle$$

$$\overset{Def.\ 1.3}{\underset{(i)}{=}}\quad(1+c_2^{-1})\frac{1}{M}\sum_{i=1}^{M}L_i^2\left\|z^{k+\frac{3}{2}}-z^{k+\frac{1}{2}}\right\|^2$$

$$+(1+c_2)\frac{1}{M}\sum_{i=1}^{M}\left\|F_i(z^{k+\frac{1}{2}})-h_i^k\right\|^2$$

$$+(1+c_2)\beta^2\omega\frac{1}{M}\sum_{i=1}^{M}\left\|F_i(z^{k+\frac{1}{2}})-h_i^k\right\|^2$$

$$-2(1+c_2)\beta\frac{1}{M}\sum_{i=1}^{M}\left\|F_i(z^{k+\frac{1}{2}})-h_i^k\right\|^2$$

$$=\quad(1+c_2^{-1})\frac{1}{M}\sum_{i=1}^{M}L_i^2\left\|z^{k+\frac{3}{2}}-z^{k+\frac{1}{2}}\right\|^2$$

$$+(1+c_2)\left[1-2\beta+\beta^2\omega\right]$$

$$\cdot \frac{1}{M} \sum_{i=1}^{M} \left\| F_i(z^{k+\frac{1}{2}}) - h_i^k \right\|^2, \tag{13}$$

where in $(i)$ we take expectation $\mathbb{E}_Q$ with respect to $Q$. Finally, we denote

$$\mathcal{Z}^k = \left\| z^k - z^* \right\|^2,$$

$$\mathcal{F}^k = \frac{1}{M} \sum_{i=1}^{M} \left\| F_i(z^{k+\frac{1}{2}}) - h_i^k \right\|^2,$$

$$\mathcal{G}^k = \left\| F(z^{k+\frac{1}{2}}) - h^k \right\|^2,$$

and sum up the estimates, obtained in equation 12, equation 13 (with coefficients $\frac{A\gamma^2}{M}$ and $B\gamma^2$, which we define later, respectively) with equation 11:

$$\mathbb{E}\mathcal{Z}^{k+1} + \frac{A\gamma^2}{M}\mathbb{E}\mathcal{F}^{k+1} + B\gamma^2\mathbb{E}\mathcal{G}^{k+1}$$

$$\leq (1-\gamma\mu)\mathbb{E}\mathcal{Z}^k + \gamma^2\frac{\omega-1}{M}\mathbb{E}\mathcal{F}^k$$

$$+\gamma^2\mathbb{E}\mathcal{G}^k - (1-\gamma\mu)\mathbb{E}\left\| z^{k+\frac{1}{2}} - z^k \right\|^2$$

$$+A\gamma^2(1+c_2^{-1})\frac{1}{M^2}\sum_{i=1}^{M} L_i^2\mathbb{E}\left\| z^{k+\frac{3}{2}} - z^{k+\frac{1}{2}} \right\|^2$$

$$+A\gamma^2(1+c_2)\left[1 - 2\beta + \beta^2\omega\right]\frac{1}{M}\mathbb{E}\mathcal{F}^k$$

$$+B\gamma^2(1+c_1^{-1})L^2\mathbb{E}\left\| z^{k+\frac{3}{2}} - z^{k+\frac{1}{2}} \right\|^2$$

$$+B\gamma^2(1+c_1)(1-\beta)^2\mathbb{E}\mathcal{G}^k$$

$$+B\gamma^2(1+c_1)\beta^2\frac{\omega-1}{M}\mathbb{E}\mathcal{F}^k$$

$$= (1-\gamma\mu)\mathbb{E}\mathcal{Z}^k$$

$$+\gamma^2\left[\frac{1}{M^2}\sum_{i=1}^{M} L_i^2 A(1+c_2^{-1}) + L^2 B(1+c_1^{-1})\right]\mathbb{E}\left\| z^{k+\frac{3}{2}} - z^{k+\frac{1}{2}} \right\|^2$$

$$+\gamma^2\left[(\omega-1) + A(1+c_2)\left[1 - 2\beta + \beta^2\omega\right] + B(1+c_1)\beta^2(\omega-1)\right]\frac{1}{M}\mathbb{E}\mathcal{F}^k$$

$$+\gamma^2\left[1 + B(1+c_1)(1-\beta)^2\right]\mathbb{E}\mathcal{G}^k$$

$$-(1-\gamma\mu)\gamma^2\mathbb{E}\left\| h^k \right\|^2. \tag{14}$$

Note that in the previous equation we used the fact, that $\left\| z^{k+\frac{1}{2}} - z^k \right\|^2 = \gamma^2 \left\| h^k \right\|^2$. Next, consider the term $\left\| z^{k+\frac{3}{2}} - z^{k+\frac{1}{2}} \right\|^2$:

$$\mathbb{E}\left\| z^{k+\frac{3}{2}} - z^{k+\frac{1}{2}} \right\|^2 = \mathbb{E}\left\| (z^{k+\frac{3}{2}} - z^{k+1}) - (z^{k+\frac{1}{2}} - z^{k+1}) \right\|^2$$

$$\overset{CB}{\leq} 2\mathbb{E}\left\| z^{k+1} - z^{k+\frac{1}{2}} \right\|^2 + 2\mathbb{E}\left\| z^{k+\frac{3}{2}} - z^{k+1} \right\|^2$$

$$= 2\gamma^2\mathbb{E}\left\| g^k - h^k \right\|^2 + 2\gamma^2\mathbb{E}\left\| h^{k+1} \right\|^2$$

$$\overset{10}{\leq} 2\gamma^2\frac{\omega-1}{M}\mathbb{E}\mathcal{F}^k + 2\gamma^2\mathbb{E}\mathcal{G}^k$$

$$+2\gamma^2\mathbb{E}\left\| h^k + \frac{1}{M}\sum_{i=1}^{M} Q\left(F_i(z^{k+\frac{1}{2}}) - h_i^k\right) \right\|^2$$

$$\overset{CB}{\leq} \quad 2\gamma^2 \frac{\omega-1}{M}\mathbb{E}\mathcal{F}^k + 2\gamma^2\mathbb{E}\mathcal{G}^k + 4\gamma^2\mathbb{E}\left\|h^k\right\|^2$$

$$+4\gamma^2\mathbb{E}\left\|\frac{1}{M}\sum_{i=1}^{M}Q\left(F_i(z^{k+\frac{1}{2}})-h_i^k\right)\right\|^2$$

$$\overset{8,9}{\leq} \quad 2\gamma^2\frac{\omega-1}{M}\mathbb{E}\mathcal{F}^k + 2\gamma^2\mathbb{E}\mathcal{G}^k$$

$$+4\gamma^2\mathbb{E}\left\|h^k\right\|^2 + 4\gamma^2\frac{\omega-1}{M}\mathbb{E}\mathcal{F}^k + 4\gamma^2\mathbb{E}\mathcal{G}^k$$

$$= \quad 6\gamma^2\frac{\omega-1}{M}\mathbb{E}\mathcal{F}^k + 6\gamma^2\mathbb{E}\mathcal{G}^k + 4\gamma^2\mathbb{E}\left\|h^k\right\|^2.$$

We substitute this estimate into equation 14:

$$\mathbb{E}\mathcal{Z}^{k+1} + \frac{A\gamma^2}{M}\mathbb{E}\mathcal{F}^{k+1} + B\gamma^2\mathbb{E}\mathcal{G}^{k+1} \quad \leq \quad (1-\gamma\mu)\mathbb{E}\mathcal{Z}^k$$

$$+\gamma^2\left[7(\omega-1) + A(1+c_2)\left[1-2\beta+\beta^2\omega\right]\right.$$

$$\left.+B(1+c_1)\beta^2(\omega-1)\right]\frac{1}{M}\mathbb{E}\mathcal{F}^k$$

$$+\gamma^2\left[7 + B(1+c_1)(1-\beta)^2\right]\mathbb{E}\mathcal{G}^k$$

$$+\gamma^2\left[4\gamma^2\left[\frac{1}{M^2}\sum_{i=1}^{M}L_i^2 A(1+c_2^{-1}) + L^2 B(1+c_1^{-1})\right]\right.$$

$$\left.-(1-\gamma\mu)\right]\mathbb{E}\left\|h^k\right\|^2. \tag{15}$$

Due to the arbitrary choice for parameters $c_1$ and $c_2$, we can set them equal to $\frac{1}{2\omega}$, and $\beta = \frac{1}{\omega}$, we instantly derive that

$$7(\omega-1) + A(1+c_2)\left[1-2\beta+\beta^2\omega\right] + B(1+c_1)\beta^2(\omega-1) \quad = \quad 7(\omega-1)$$

$$+ \quad A\left(1+\frac{1}{2\omega}\right)\left(1-\frac{1}{\omega}\right)$$

$$+ \quad B\left(1+\frac{1}{2\omega}\right)\left(1-\frac{1}{\omega}\right)$$

$$\leq \quad 7(\omega-1)$$

$$+ \quad A\left(1-\frac{1}{2\omega}\right)$$

$$+ \quad B\left(1-\frac{1}{2\omega}\right),$$

and also

$$7 + B(1+c_1)(1-\beta)^2 \quad = \quad 7 + B\left(1+\frac{1}{2\omega}\right)\left(1-\frac{1}{\omega}\right)^2$$

$$\leq \quad 7 + B\left(1-\frac{1}{2\omega}\right).$$

Summing up equation 15 with $p^k = (1-\gamma\mu)^{-k}$ and substituting the obtained estimations,

$$\sum_{k=0}^{K-1}p^k\mathbb{E}\mathcal{Z}^{k+1} + \frac{A\gamma^2}{M}\sum_{k=0}^{K-1}p^k\mathbb{E}\mathcal{F}^{k+1} \quad + \quad B\gamma^2\sum_{k=0}^{K-1}p^k\mathbb{E}\mathcal{G}^{k+1}$$

$$\leq \quad \sum_{k=0}^{K-1}p^k(1-\gamma\mu)\mathbb{E}\mathcal{Z}^k$$

$$+\gamma^2\left[7(\omega-1) + A\left(1-\frac{1}{2\omega}\right) + B\left(1-\frac{1}{2\omega}\right)\right]$$

$$\cdot \frac{1}{M} \sum_{k=0}^{K-1} p^k \mathbb{E} \mathcal{F}^k$$

$$+ \gamma^2 \left[ 7 + B \left( 1 - \frac{1}{2\omega} \right) \right]$$

$$\cdot \sum_{k=0}^{K-1} p^k \mathbb{E} \mathcal{G}^k$$

$$+ \gamma^2 \left[ 4\gamma^2 \left[ \frac{1}{M^2} \sum_{i=1}^{M} L_i^2 A(1 + 2\omega) + L^2 B(1 + 2\omega) \right] \right.$$

$$\left. - (1 - \gamma\mu) \right] \sum_{k=0}^{K-1} p^k \mathbb{E} \left\| h^k \right\|^2.$$

Next, we derive the following estimates with $p^{-1} = 1 - \gamma\mu$:

$$p \left[ 7(\omega - 1) + A \left( 1 - \frac{1}{2\omega} \right) + B \left( 1 - \frac{1}{2\omega} \right) \right] = \frac{7(\omega - 1) + A \left( 1 - \frac{1}{2\omega} \right) + B \left( 1 - \frac{1}{2\omega} \right)}{1 - \gamma\mu}$$

$$\leq \frac{7(\omega - 1) + A \left( 1 - \frac{1}{2\omega} \right) + B \left( 1 - \frac{1}{2\omega} \right)}{1 - \frac{1}{4\omega}}$$

$$\leq 10(\omega - 1) + A \left( 1 - \frac{1}{4\omega} \right) + B \left( 1 - \frac{1}{4\omega} \right).$$

In order to fulfill the last inequality, we choose $\gamma \leq \frac{1}{4\omega\mu}$. One can also note, that

$$p \left[ 7 + B \left( 1 - \frac{1}{2\omega} \right) \right] = \frac{7 + B \left( 1 - \frac{1}{2\omega} \right)}{1 - \gamma\mu}$$

$$\leq \frac{7 + B \left( 1 - \frac{1}{2\omega} \right)}{1 - \frac{1}{4\omega}}$$

$$\leq 10 + B \left( 1 - \frac{1}{4\omega} \right).$$

We choose $A$ and $B$ to satisfy

$$\begin{cases} A \geq 10(\omega - 1) + A \left( 1 - \frac{1}{4\omega} \right) + B \left( 1 - \frac{1}{4\omega} \right), \\ B \geq 10 + B \left( 1 - \frac{1}{4\omega} \right), \end{cases}$$

$$\begin{cases} A \geq 200\omega^2, \\ B \geq 40\omega. \end{cases}$$

If we choose

$$\gamma \leq \min \left\{ \frac{1}{4\omega\mu}, \frac{1}{8\sqrt{30}\omega L}, \frac{\sqrt{M}}{40\sqrt{6} \cdot \omega^{3/2} \sqrt{\frac{1}{M} \sum_{i=1}^{M} L_i^2}} \right\},$$

then it is obvious, that

$$4\gamma^2 \frac{1}{M^2} \sum_{i=1}^{M} L_i^2 A(1 + 2\omega) + 4\gamma^2 L^2 B(1 + 2\omega) - (1 - \gamma\mu) \leq \frac{1}{2} - (1 - \gamma\mu) < 0.$$

Lastly, we get

$$\sum_{k=0}^{K-1} p^k \mathbb{E} \mathcal{Z}^{k+1} + \sum_{k=0}^{K-1} p^k \frac{A\gamma^2}{M} \mathbb{E} \mathcal{F}^{k+1} + \sum_{k=0}^{K-1} p^k B\gamma^2 \mathbb{E} \mathcal{G}^{k+1} \leq \sum_{k=0}^{K-1} p^k (1 - \gamma\mu) \mathbb{E} \mathcal{Z}^k$$

$$+ \sum_{k=0}^{K-1} p^{k-1} \frac{A\gamma^2}{M} \mathbb{E} \mathcal{F}^k$$

$$+ \sum_{k=0}^{K-1} p^{k-1} \gamma^2 B \mathbb{E} \mathcal{G}^k.$$

Using the choice of $p$ we derive the final estimate:

$$\sum_{k=0}^{K-1}(1-\gamma\mu)^{-k}\mathbb{E}\mathcal{Z}^{k+1} + \sum_{k=0}^{K-1}(1-\gamma\mu)^{-k}\frac{A\gamma^2}{M}\mathbb{E}\mathcal{F}^{k+1} + \sum_{k=0}^{K-1}(1-\gamma\mu)^{-k}B\gamma^2\mathbb{E}\mathcal{G}^{k+1}$$

$$\leq \sum_{k=0}^{K-1}(1-\gamma\mu)^{-k+1}\mathbb{E}\mathcal{Z}^k$$

$$+ \sum_{k=0}^{K-1}(1-\gamma\mu)^{-k+1}\frac{A\gamma^2}{M}\mathbb{E}\mathcal{F}^k$$

$$+ \sum_{k=0}^{K-1}(1-\gamma\mu)^{-k+1}B\gamma^2\mathbb{E}\mathcal{G}^k;$$

$$(1-\gamma\mu)^{-K+1}\mathbb{E}\mathcal{Z}^K + (1-\gamma\mu)^{-K+1}\frac{A\gamma^2}{M}\mathbb{E}\mathcal{F}^K + (1-\gamma\mu)^{-K+1}B\gamma^2\mathbb{E}\mathcal{G}^K$$

$$\leq (1-\gamma\mu)\mathcal{Z}^0$$

$$+(1-\gamma\mu)\gamma^2\frac{A\gamma^2}{M}\mathcal{F}^0$$

$$+(1-\gamma\mu)\gamma^2\mathcal{G}^0;$$

$$\mathbb{E}V^K \leq (1-\gamma\mu)^K\mathbb{E}V^0,$$

where we denote $V^k = \mathcal{Z}^k + \frac{A\gamma^2}{M}\mathcal{F}^k + B\gamma^2\mathcal{G}^k$. This ends the proof of the theorem.

$\square$

**Corollary 2.6** *Suppose the conditions of Theorem 2.5 holds. Then Algorithm 1 has*

$$\mathcal{O}\left(\left(\frac{\omega}{q_\omega} + \frac{\omega L}{\mu q_\omega} + \frac{\omega^{3/2}\hat{L}}{\mu q_\omega \sqrt{M}}\right)\log\left(\frac{1}{\varepsilon}\right)\right)$$

*communication complexity, where $q_\omega$ is the expected density and $\varepsilon$ represents accuracy of the solution.*

*Proof.* Having result of Theorem 2.5, we substitute the choice of $\gamma$ and obtain the following iteration complexity:

$$\mathcal{O}\left(\left(\omega + \frac{\omega L}{\mu} + \frac{\omega^{3/2}\hat{L}}{\mu\sqrt{M}}\right)\log\left(\frac{1}{\varepsilon}\right)\right).$$

Now assuming, that at each iteration we transmit $\frac{1}{q_\omega}$ bits of information, where $q_\omega$ is the expected density, we ends the proof of the corollary.

$\square$

## D   PROOF OF THEOREM 2.8

**Theorem 2.8** *Suppose Assumptions 2.1, 2.3 hold. Then for Algorithm 2 with $\gamma \leq$*

$$\min\left\{\frac{\alpha}{8\mu}, \frac{\alpha^2}{16\sqrt{1772}\sqrt{\frac{1}{M}\sum_{i=1}^{M}L_i^2}}\right\} \text{ and } \beta = 1 - \frac{\alpha}{8} \text{ the following estimate is fulfilled:}$$

$$\mathbb{E}V^K \leq (1-\gamma\mu)^K V^0,$$

*where $V^k = \left\|z^k - z^*\right\|^2 + \frac{7088\gamma^2}{\alpha^3}\frac{1}{M}\sum_{i=1}^{M}\left\|F_i(z^{k+\frac{1}{2}}) - h_i^k\right\|^2 + \frac{276\gamma^2}{\alpha}\frac{1}{M}\sum_{i=1}^{M}\left\|e_i^k\right\|.$*

*Proof.* Firstly, we introduce new sequences $\hat{z}^k$ and $\hat{z}^{k+\frac{1}{2}}$:

$$\hat{z}^k := z^k - \gamma e^k, \tag{16}$$

$$\hat{z}^{k+\frac{1}{2}} := z^{k+\frac{1}{2}} - \gamma e^k. \tag{17}$$

Note then, that the following equal transition is held:

$$\hat{z}^{k+1} \overset{16}{=} z^{k+1} - \gamma e^{k+1}$$

$$= (z^k - \gamma g^k) - \gamma \frac{1}{M} \sum_{i=1}^{M} \left[ e_i^k + \Delta_i^k - \hat{\Delta}_i^k \right]$$

$$= z^k - \gamma e^k - \gamma g^k - \gamma \frac{1}{M} \sum_{i=1}^{M} \left[ F_i^k(z^{k+\frac{1}{2}}) - h_i^k + h_i^k - g_i^k \right]$$

$$= \hat{z}^k - \gamma F(z^{k+\frac{1}{2}}).$$

We use a classical for EXTRAGRADIENT proofs equation for arbitrary $z, v$ and $u$:

$$\|z - u\|^2 = \|v - u\|^2 + 2 \langle z - v, z - u \rangle - \|z - v\|^2.$$

For $z = z^{k+\frac{1}{2}}$, $u = z^*$ and $v = \hat{z}^k$:

$$\left\| z^{k+\frac{1}{2}} - z^* \right\|^2 = \left\| \hat{z}^k - z^* \right\|^2 + 2 \left\langle z^{k+\frac{1}{2}} - \hat{z}^k, z^{k+\frac{1}{2}} - z^* \right\rangle - \left\| z^{k+\frac{1}{2}} - \hat{z}^k \right\|^2.$$

Using the same for $z = \hat{z}^{k+1}$ with $u = z^*$ and $v = z^{k+\frac{1}{2}}$,

$$\left\| \hat{z}^{k+1} - z^* \right\|^2 = \left\| z^{k+\frac{1}{2}} - z^* \right\|^2 + 2 \left\langle \hat{z}^{k+1} - z^{k+\frac{1}{2}}, z^{k+\frac{1}{2}} - z^* \right\rangle + \left\| z^{k+\frac{1}{2}} - \hat{z}^{k+1} \right\|^2.$$

After the summation with the previous equality, we obtain

$$\left\| \hat{z}^{k+1} - z^* \right\|^2 = \left\| \hat{z}^k - z^* \right\|^2 + 2 \left\langle \hat{z}^{k+1} - \hat{z}^k, z^{k+\frac{1}{2}} - z^* \right\rangle$$

$$+ \left\| z^{k+\frac{1}{2}} - \hat{z}^{k+1} \right\|^2 - \left\| z^{k+\frac{1}{2}} - \hat{z}^k \right\|^2. \tag{18}$$

Let's work with the term $\left\| \hat{z}^{k+1} - z^{k+\frac{1}{2}} \right\|^2$:

$$\left\| \hat{z}^{k+1} - z^{k+\frac{1}{2}} \right\|^2 = 2 \left\| \hat{z}^{k+1} - \hat{z}^{k+\frac{1}{2}} \right\|^2 + 2 \left\| \hat{z}^{k+\frac{1}{2}} - z^{k+\frac{1}{2}} \right\|^2$$

$$= 2\gamma^2 \left\| e^{k+1} - e^k + g^k - h^k \right\|^2 + 2\gamma^2 \left\| e^k \right\|^2$$

$$\overset{CB}{\leq} 4\gamma^2 \left\| \frac{1}{M} \sum_{i=1}^{M} \left[ \Delta_i^k - \hat{\Delta}_i^k \right] \right\|^2 + 4\gamma^2 \left\| g^k - h^k \right\| + 2\gamma^2 \left\| e^k \right\|^2$$

$$\overset{CB}{\leq} 8\gamma^2$$

$$\cdot \left\| \frac{1}{M} \sum_{i=1}^{M} \left[ \left( F_i(z^{k+\frac{1}{2}}) - h_i^k + e_i^k \right) - Q \left( F_i(z^{k+\frac{1}{2}}) - h_i^k + e_i^k \right) \right] \right\|^2$$

$$+ 8\gamma^2 \left\| e^k \right\|^2 + 4\gamma^2 \left\| g^k - h^k \right\| + 2\gamma^2 \left\| e^k \right\|^2$$

$$\overset{\text{Def. 1.5}}{\underset{CB}{\leq}} 8\gamma^2(1-\alpha) \frac{1}{M} \sum_{i=1}^{M} \left\| F_i(z^{k+\frac{1}{2}}) - h_i^k + e_i^k \right\|^2 + 10\gamma^2 \left\| e^k \right\|^2$$

$$+ 4\gamma^2 \left\| g^k - h^k \right\|^2$$

$$\overset{CB}{\leq} 16\gamma^2(1-\alpha) \frac{1}{M} \sum_{i=1}^{M} \left\| F_i(z^{k+\frac{1}{2}}) - h_i^k \right\|^2$$

$$+ (26 - 16\alpha)\gamma^2 \frac{1}{M} \sum_{i=1}^{M} \left\| e_i^k \right\|^2 + 4\gamma^2 \left\| g^k - h^k \right\|^2. \tag{19}$$

The term $2 \left\langle \hat{z}^{k+1} - \hat{z}^k, z^{k+\frac{1}{2}} - z^* \right\rangle$ can be bounded in the following way, using the definition of $\hat{z}^{k+1}$ and $\mu$-strongly monotonicity of operator $F$:

$$2 \left\langle \hat{z}^{k+1} - \hat{z}^k, z^{k+\frac{1}{2}} - z^* \right\rangle \overset{16}{=} -2\gamma \left\langle F(z^{k+\frac{1}{2}}), z^{k+\frac{1}{2}} - z^* \right\rangle$$

$$= -2\gamma \left\langle F(z^{k+\frac{1}{2}}) - F(z^*), z^{k+\frac{1}{2}} - z^* \right\rangle$$

$$-2\gamma \underbrace{\left\langle F(z^*), z^{k+\frac{1}{2}} - z^* \right\rangle}_{\leq 0}$$

$$\overset{Ass.\ 2.3}{\leq} -2\gamma\mu \left\| z^{k+\frac{1}{2}} - z^* \right\|^2 .$$

Substituting this result with equation 19 into equation 18 we get

$$
\begin{aligned}
\left\| \hat{z}^{k+1} - z^* \right\|^2 \quad \leq \quad & \left\| \hat{z}^k - z^* \right\|^2 - 2\gamma\mu \left\| z^{k+\frac{1}{2}} - z^* \right\|^2 \\
& + 16\gamma^2 (1-\alpha) \frac{1}{M} \sum_{i=1}^{M} \left\| F_i(z^{k+\frac{1}{2}}) - h_i^k \right\|^2 \\
& + (26 - 16\alpha)\gamma^2 \frac{1}{M} \sum_{i=1}^{M} \left\| e_i^k \right\|^2 \\
& + 4\gamma^2 \left\| g^k - h^k \right\|^2 - \left\| z^{k+\frac{1}{2}} - \hat{z}^k \right\|^2 \\
= \quad & (1-\gamma\mu) \left\| \hat{z}^k - z^* \right\|^2 \\
& + 16\gamma^2 (1-\alpha) \frac{1}{M} \sum_{i=1}^{M} \left\| F_i(z^{k+\frac{1}{2}}) - h_i^k \right\|^2 \\
& + \gamma^2 (26 - 16\alpha) \frac{1}{M} \sum_{i=1}^{M} \left\| e_i^k \right\|^2 \\
& + 4\gamma^2 \left\| g^k - h^k \right\|^2 - (1-\gamma\mu) \left\| z^{k+\frac{1}{2}} - \hat{z}^k \right\|^2 \\
\overset{Jensen}{\leq} \quad & (1-\gamma\mu) \left\| \hat{z}^k - z^* \right\|^2 \\
& + 16\gamma^2 (1-\alpha) \frac{1}{M} \sum_{i=1}^{M} \left\| F_i(z^{k+\frac{1}{2}}) - h_i^k \right\|^2 \\
& + \gamma^2 (27 - 16\alpha - \gamma\mu) \frac{1}{M} \sum_{i=1}^{M} \left\| e_i^k \right\|^2 \\
& + 4\gamma^2 \left\| g^k - h^k \right\|^2 - \frac{1-\gamma\mu}{2} \left\| z^{k+\frac{1}{2}} - z^k \right\|^2 .
\end{aligned}
$$

Once again using the Jensen inequality, we deal with the term $\left\| g^k - h^k \right\|^2$:

$$
\begin{aligned}
\left\| g^k - h^k \right\|^2 \quad = \quad & \left\| \hat{\Delta}^k \right\| = \left\| \frac{1}{M} \sum_{i=1}^{M} \hat{\Delta}_i^k \right\|^2 \\
\overset{CB}{\leq} \quad & \frac{2}{M} \sum_{i=1}^{M} \left\| Q\left( F_i(z^{k+\frac{1}{2}}) - h_i^k + e_i^k \right) - \left( F_i(z^{k+\frac{1}{2}}) - h_i^k + e_i^k \right) \right\|^2 \\
& + \frac{2}{M} \sum_{i=1}^{M} \left\| F_i(z^{k+\frac{1}{2}}) - h_i^k + e_i^k \right\|^2 \\
\overset{Def.\ 1.5}{\leq} \quad & \frac{4 - 2\alpha}{M} \sum_{i=1}^{M} \left\| F_i(z^{k+\frac{1}{2}}) - h_i^k + e_i^k \right\|^2 \\
\overset{CB}{\leq} \quad & \frac{8 - 4\alpha}{M} \sum_{i=1}^{M} \left\| F_i(z^{k+\frac{1}{2}}) - h_i^k \right\|^2 + \frac{8 - 4\alpha}{M} \sum_{i=1}^{M} \left\| e_i^k \right\|^2 . \qquad (20)
\end{aligned}
$$

In that way, we achieve

$$\left\| \hat{z}^{k+1} - z^* \right\|^2 \quad \leq \quad (1-\gamma\mu) \left\| \hat{z}^k - z^* \right\|^2$$

$$+\gamma^2(48-32\alpha)\frac{1}{M}\sum_{i=1}^{M}\left\|F_i(z^{k+\frac{1}{2}})-h_i^k\right\|^2$$

$$+\gamma^2\left(59-32\alpha\right)\frac{1}{M}\sum_{i=1}^{M}\left\|e_i^k\right\|^2$$

$$-\frac{1-\gamma\mu}{2}\left\|z^{k+\frac{1}{2}}-z^k\right\|^2. \tag{21}$$

Let's derive the recursion on the terms $\frac{1}{M}\sum_{i=1}^{M}\left\|F_i(z^{k+\frac{3}{2}})-h_i^{k+1}\right\|^2$ and $\frac{1}{M}\sum_{i=1}^{M}\left\|e_i^{k+1}\right\|^2$. Firstly, one can note that

$$\frac{1}{M}\sum_{i=1}^{M}\left\|F_i(z^{k+\frac{3}{2}})-h_i^{k+1}\right\|^2$$

$$=\quad\frac{1}{M}\sum_{i=1}^{M}\left\|\left(F_i(z^{k+\frac{3}{2}})-F_i(z^{k+\frac{1}{2}})\right)+\left(F_i(z^{k+\frac{1}{2}})-h_i^{k+1}\right)\right\|^2$$

$$\overset{Jensen}{\leq}\quad(1+s_1^{-1})\frac{1}{M}\sum_{i=1}^{M}\left\|F_i(z^{k+\frac{3}{2}})-F_i(z^{k+\frac{1}{2}})\right\|^2$$

$$+(1+s_1)\frac{1}{M}\sum_{i=1}^{M}\left\|F_i(z^{k+\frac{1}{2}})-h_i^{k+1}\right\|^2$$

$$\overset{Ass.\ 2.1}{\leq}\quad(1+s_1^{-1})\frac{1}{M}\sum_{i=1}^{M}L_i^2\left\|z^{k+\frac{3}{2}}-z^{k+\frac{1}{2}}\right\|^2$$

$$+(1+s_1)\frac{1}{M}\sum_{i=1}^{M}\left\|F_i(z^{k+\frac{1}{2}})-h_i^k-\beta Q(F_i(z^{k+\frac{1}{2}})-h_i^k)\right\|^2$$

$$\overset{Jensen}{\leq}\quad(1+s_1^{-1})\frac{1}{M}\sum_{i=1}^{M}L_i^2\left\|z^{k+\frac{3}{2}}-z^{k+\frac{1}{2}}\right\|^2$$

$$+(1+s_1)(1+s_2)\frac{1}{M}\sum_{i=1}^{M}\left\|F_i(z^{k+\frac{1}{2}})-h_i^k-Q(F_i(z^{k+\frac{1}{2}})-h_i^k)\right\|^2$$

$$+(1+s_1)(1+s_2^{-1})(1-\beta)^2\frac{1}{M}\sum_{i=1}^{M}\left\|Q\left(F_i(z^{k+\frac{1}{2}})-h_i^k\right)\right\|^2$$

$$\overset{Def.\ 1.5}{\underset{CB}{\leq}}\quad(1+s_1^{-1})\frac{1}{M}\sum_{i=1}^{M}L_i^2\left\|z^{k+\frac{3}{2}}-z^{k+\frac{1}{2}}\right\|^2$$

$$+(1+s_1)(1+s_2)(1-\alpha)\frac{1}{M}\sum_{i=1}^{M}\left\|F_i(z^{k+\frac{1}{2}})-h_i^k\right\|^2$$

$$+2(1+s_1)(1+s_2^{-1})(1-\beta)^2\frac{1}{M}\sum_{i=1}^{M}\left\|Q\left(F_i(z^{k+\frac{1}{2}})-h_i^k\right)-\left(F_i(z^{k+\frac{1}{2}})-h_i^k\right)\right\|^2$$

$$+2(1+s_1)(1+s_2^{-1})(1-\beta)^2\frac{1}{M}\sum_{i=1}^{M}\left\|F_i(z^{k+\frac{1}{2}})-h_i^k\right\|^2$$

$$\overset{Def.\ 1.5}{\leq}\quad(1+s_1^{-1})\frac{1}{M}\sum_{i=1}^{M}L_i^2\left\|z^{k+\frac{3}{2}}-z^{k+\frac{1}{2}}\right\|^2$$

$$+(1+s_1)(1+s_2)(1-\alpha)\frac{1}{M}\sum_{i=1}^{M}\left\|F_i(z^{k+\frac{1}{2}})-h_i^k\right\|^2$$

$$+(1+s_1)(1+s_2^{-1})(1-\beta)^2(4-2\alpha)\frac{1}{M}\sum_{i=1}^{M}\left\|F_i(z^{k+\frac{1}{2}})-h_i^k\right\|^2. \tag{22}$$

Now we proceed to $\frac{1}{M}\sum_{i=1}^{M}\left\|e_i^{k+1}\right\|^2$:

$$
\begin{aligned}
\frac{1}{M}\sum_{i=1}^{M}\left\|e_i^{k+1}\right\|^2 &= \frac{1}{M}\sum_{i=1}^{M}\left\|e_i^k+\Delta_i^k-\hat{\Delta}_i^k\right\|^2 \\
&= \frac{1}{M}\sum_{i=1}^{M}\left\|F_i(z^{k+\frac{1}{2}})-h_i^k+e_i^k-Q\left(F_i(z^{k+\frac{1}{2}})-h_i^k+e_i^k\right)\right\|^2 \\
&\overset{\substack{\text{Def. 1.5}\\ Jensen}}{\leq} (1+s_3^{-1})(1-\alpha)\frac{1}{M}\sum_{i=1}^{M}\left\|F_i(z^{k+\frac{1}{2}})-h_i^k\right\|^2
\end{aligned}
$$

$$+(1+s_3)(1-\alpha)\frac{1}{M}\sum_{i=1}^{M}\left\|e_i^k\right\|^2. \tag{23}$$

Finally, we denote

$$
\begin{aligned}
\mathcal{Z}^k &= \left\|\hat{z}^{k+1}-z^*\right\|^2, \\
\mathcal{F}^k &= \frac{1}{M}\sum_{i=1}^{M}\left\|F_i(z^{k+\frac{1}{2}})-h_i^k\right\|^2, \\
\mathbb{E}^k &= \frac{1}{M}\sum_{i=1}^{M}\left\|e_i^k\right\|^2
\end{aligned}
$$

and sum up the estimates, obtained in equation 22, equation 23 (with coefficients $A\gamma^2$ and $C\gamma^2$, which we define later, respectively) with equation 21:

$$
\begin{aligned}
\mathcal{Z}^{k+1}+A\gamma^2\mathcal{F}^{k+1}+C\gamma^2\mathbb{E}^{k+1} \leq\; &(1-\gamma\mu)\mathcal{Z}^k \\
&+\gamma^2(48-32\alpha)\mathcal{F}^k \\
&+\gamma^2(59-32\alpha)\mathbb{E}^k \\
&-\frac{1-\gamma\mu}{2}\left\|z^{k+\frac{1}{2}}-z^k\right\|^2 \\
&+A\gamma^2(1+s_1^{-1})\frac{1}{M}\sum_{i=1}^{M}L_i^2\left\|z^{k+\frac{3}{2}}-z^{k+\frac{1}{2}}\right\|^2 \\
&+A\gamma^2(1+s_1)(1+s_2)(1-\alpha)\mathcal{F}^k \\
&+A\gamma^2(1+s_1)(1+s_2^{-1})(1-\beta)^2(4-2\alpha)\mathcal{F}^k \\
&+C\gamma^2(1+s_3^{-1})(1-\alpha)\mathcal{F}^k \\
&+C\gamma^2(1+s_3)(1-\alpha)\mathbb{E}^k \\
=\; &(1-\gamma\mu)\mathcal{Z}^k \\
&+\gamma^2\left[(48-32\alpha)+A(1+s_1)(1+s_2)(1-\alpha)\right. \\
&\quad+A(1+s_1)(1+s_2^{-1})(1-\beta)^2(4-2\alpha) \\
&\quad\left.+C(1+s_3^{-1})(1-\alpha)\right]\mathcal{F}^k \\
&+\gamma^2\left[(59-32\alpha)+C(1+s_3)(1-\alpha)\right]\mathbb{E}^k \\
&-\frac{1-\gamma\mu}{2}\gamma^2\left\|h^k\right\|^2 \\
&+A\gamma^2(1+s_1^{-1})\frac{1}{M}\sum_{i=1}^{M}L_i^2\left\|z^{k+\frac{3}{2}}-z^{k+\frac{1}{2}}\right\|^2. \tag{24}
\end{aligned}
$$

Note that in the previous equation we used the fact, that $\left\| z^{k+\frac{1}{2}} - z^k \right\|^2 = \gamma^2 \left\| h^k \right\|^2$. Next, consider the last term:

$$
\begin{aligned}
\left\| z^{k+\frac{3}{2}} - z^{k+\frac{1}{2}} \right\|^2 &= \left\| z^{k+\frac{3}{2}} - z^{k+\frac{1}{2}} \pm z^{k+1} \right\|^2 \\
&\overset{CB}{\leq} 2 \left\| z^{k+1} - z^{k+\frac{1}{2}} \right\|^2 + 2 \left\| z^{k+\frac{3}{2}} - z^{k+1} \right\|^2 \\
&= 2\gamma^2 \left\| g^k - h^k \right\|^2 + 2\gamma^2 \left\| h^{k+1} \right\|^2 \\
&\overset{20}{\leq} (16 - 8\alpha)\gamma^2 \mathcal{F}^k + (16 - 8\alpha)\gamma^2 \mathbb{E}^k + 2\gamma^2 \left\| h^{k+1} \right\|^2 \\
&\overset{CB}{\leq} (16 - 8\alpha)\gamma^2 \mathcal{F}^k + (16 - 8\alpha)\gamma^2 \mathbb{E}^k \\
&\quad + 2\gamma^2 \left\| h^k + \beta \frac{1}{M} \sum_{i=1}^{M} Q\left( F_i(z^{k+\frac{1}{2}}) - h_i^k \right) \right\|^2 \\
&\overset{CB}{\leq} (16 - 8\alpha)\gamma^2 \mathcal{F}^k + (16 - 8\alpha)\gamma^2 \mathbb{E}^k \\
&\quad + 4\gamma^2 \left\| h^k \right\|^2 + (16 - 8\alpha)\gamma^2 \beta^2 \mathcal{F}^k,
\end{aligned}
$$

which we substitute this term into the equation 24:

$$
\begin{aligned}
\mathcal{Z}^{k+1} + A\gamma^2 \mathcal{F}^{k+1} + C\gamma^2 \mathbb{E}^{k+1} \leq\ & (1 - \gamma\mu)\mathcal{Z}^k \\
&+ \gamma^2 \Big[ (48 - 32\alpha) + A(1 + s_1)(1 + s_2)(1 - \alpha) \\
&\quad + A(1 + s_1^{-1})(16 - 8\alpha)(1 + \beta^2)\gamma^2 \frac{1}{M} \sum_{i=1}^{M} L_i^2 \\
&\quad + A(1 + s_1)(1 + s_2^{-1})(1 - \beta)^2(4 - 2\alpha) \\
&\quad + C(1 + s_3^{-1})(1 - \alpha) \Big] \mathcal{F}^k \\
&+ \gamma^2 \Big[ (59 - 32\alpha) + C(1 + s_3)(1 - \alpha) \\
&\quad + A(1 + s_1^{-1})\gamma^2 \frac{1}{M} \sum_{i=1}^{M} L_i^2(16 - 8\alpha) \Big] \mathbb{E}^k \\
&+ \gamma^2 \left[ 4A(1 + s_1^{-1})\gamma^2 \frac{1}{M} \sum_{i=1}^{M} L_i^2 - \frac{1 - \gamma\mu}{2} \right] \left\| h^k \right\|^2.
\end{aligned}
$$

Due to the arbitrary choice for parameters $s_1$ and $s_2$, we can set them equal to $\frac{1}{\sqrt[4]{1-\alpha}} - 1$, from which instantly follows that

$$
1 + s_{1,2} = \frac{1}{\sqrt[4]{1-\alpha}} \qquad \text{and} \qquad 1 + s_{1,2}^{-1} = \frac{1}{1 - \sqrt[4]{1-\alpha}} \leq \frac{4}{\alpha},
$$

similarly, for products:

$$
\begin{aligned}
(1 + s_1)(1 + s_2) &= \frac{1}{\sqrt{1-\alpha}}, \\
(1 + s_1)(1 + s_2^{-1}) &= \frac{1}{\sqrt[4]{1-\alpha} \cdot \left(1 - \sqrt[4]{1-\alpha}\right)} \leq \frac{4}{\alpha}.
\end{aligned}
$$

Inserting these results into the coefficient in front of $\mathcal{F}^k$ and assuming $s_3^{-1} = \frac{2}{\alpha}$, one can obtain:

$$
\begin{aligned}
(48 - 32\alpha)\ &+\ A(1 + s_1)(1 + s_2)(1 - \alpha) \\
&+\ A(1 + s_1^{-1})(16 - 8\alpha)(1 + \beta^2 \gamma^2 \frac{1}{M} \sum_{i=1}^{M} L_i^2 \\
&+\ A(1 + s_1)(1 + s_2^{-1})(1 - \beta)^2(4 - 2\alpha) \\
&+\ C(1 + s_3^{-1})(1 - \alpha) \\
&=\ (48 - 32\alpha) + A\sqrt{1-\alpha}
\end{aligned}
$$

$$+ \quad A \left( \frac{1}{1 - \sqrt[4]{1-\alpha}} \right) (16 - 8\alpha)(1 + \beta^2)\gamma^2 \frac{1}{M} \sum_{i=1}^{M} L_i^2$$

$$+ \quad A \frac{1}{\sqrt[4]{1-\alpha} \cdot \left(1 - \sqrt[4]{1-\alpha}\right)} (1 - \beta)^2 (4 - 2\alpha) + C \left(1 + \frac{2}{\alpha}\right)(1 - \alpha)$$

$$\leq \quad (48 - 32\alpha) + A \left(1 - \frac{\alpha}{2}\right) + 16A \left(\frac{4}{\alpha}\right)\left(1 - \frac{\alpha}{2}\right)(1 + \beta^2)\gamma^2 \frac{1}{M} \sum_{i=1}^{M} L_i^2$$

$$+ \quad A \cdot \left(\frac{4}{\alpha}\right)(1 - \beta)^2 (4 - 2\alpha) + C \cdot \frac{2}{\alpha}$$

Consider $\beta = 1 - \frac{\alpha}{8}$, therefore

$$(48 - 32\alpha) + A \left(1 - \frac{\alpha}{2}\right) \quad + \quad 16A \left(\frac{4}{\alpha}\right)\left(1 - \frac{\alpha}{2}\right)(1 + \beta^2)\gamma^2 \frac{1}{M} \sum_{i=1}^{M} L_i^2$$

$$+ \quad A \left(\frac{4}{\alpha}\right)(1 - \beta)^2 (4 - 2\alpha) + C \cdot \frac{2}{\alpha}$$

$$\leq \quad (48 - 32\alpha) + A \left(1 - \frac{\alpha}{2}\right) + 128 \frac{A}{\alpha}\gamma^2 \frac{1}{M} \sum_{i=1}^{M} L_i^2$$

$$+ \quad A \cdot \frac{\alpha}{4} + C \cdot \frac{2}{\alpha}.$$

And finally, if $\gamma \leq \dfrac{1}{8\sqrt{\frac{1}{M} \sum\limits_{i=1}^{M} L_i^2}} \sqrt{\frac{\alpha}{A}}$ we get

$$(48 - 32\alpha) + A \left(1 - \frac{\alpha}{2}\right) + 2 + A \cdot \frac{\alpha}{4} + C \cdot \frac{2}{\alpha} = (50 - 32\alpha) + A \left(1 - \frac{\alpha}{4}\right) + C \cdot \frac{2}{\alpha}.$$

Further, we do the similar thing for the coefficient before $\mathbb{E}^k$:

$$(59 - 32\alpha) + C(1 + s_3)(1 - \alpha) \quad + \quad A(1 + s_1^{-1})\gamma^2 \frac{1}{M} \sum_{i=1}^{M} L_i^2 (16 - 8\alpha)$$

$$\leq \quad (59 - 32\alpha) + C \left(1 + \frac{\alpha}{2}\right)(1 - \alpha)$$

$$+ \quad A \frac{64}{\alpha}\gamma^2 \frac{1}{M} \sum_{i=1}^{M} L_i^2 \left(1 - \frac{\alpha}{2}\right)$$

$$\leq \quad (60 - 32\alpha) + C \left(1 - \frac{\alpha}{2}\right).$$

Summing with coefficients $p^k = (1 - \gamma\mu)^{-k}$:

$$\sum_{k=0}^{K-1} p^k \mathcal{Z}^{k+1} + \sum_{k=0}^{K-1} p^k A\gamma^2 \mathcal{F}^{k+1} + \sum_{k=0}^{K-1} p^k C\gamma^2 \mathbb{E}^{k+1}$$

$$\leq \sum_{k=0}^{K-1} p^k (1 - \gamma\mu)\mathcal{Z}^k$$

$$+ \sum_{k=0}^{K-1} p^k \gamma^2 \left[(50 - 32\alpha) + A \left(1 - \frac{\alpha}{4}\right) + C \cdot \frac{2}{\alpha}\right] \mathcal{F}^k$$

$$+ \sum_{k=0}^{K-1} p^k \gamma^2 \left[(60 - 32\alpha) + C \left(1 - \frac{\alpha}{2}\right)\right] \mathbb{E}^k$$

$$+ \sum_{k=0}^{K-1} p^k \gamma^2 \left[\frac{16A}{\alpha}\gamma^2 \frac{1}{M} \sum_{i=1}^{M} L_i^2 - \frac{1 - \gamma\mu}{2}\right] \|h^k\|^2.$$

Now we found the following estimates with $p^{-1} = 1 - \gamma\mu$:

$$p\left[(50 - 32\alpha) + A\left(1 - \frac{\alpha}{4}\right) + C \cdot \frac{2}{\alpha}\right] = \frac{(50 - 32\alpha) + A\left(1 - \frac{\alpha}{4}\right) + C \cdot \frac{2}{\alpha}}{1 - \gamma\mu}.$$

Now we choose $\gamma \leq \frac{\alpha}{8\mu}$ to fulfill next inequality:

$$\frac{1}{1 - \gamma\mu} \leq \frac{1}{1 - \frac{\alpha}{8}}.$$

In that way,

$$\frac{(50 - 32\alpha) + A\left(1 - \frac{\alpha}{4}\right) + C \cdot \frac{2}{\alpha}}{1 - \gamma\mu} \leq \frac{(50 - 32\alpha) + A\left(1 - \frac{\alpha}{4}\right) + C \cdot \frac{2}{\alpha}}{1 - \frac{\alpha}{8}}$$

$$\leq (58 - 37\alpha) + A\left(1 - \frac{\alpha}{8}\right) + C \cdot \frac{3}{\alpha}.$$

Analogically, we can estimate

$$p\left[(60 - 32\alpha) + C\left(1 - \frac{\alpha}{2}\right)\right] = \frac{(60 - 32\alpha) + C\left(1 - \frac{\alpha}{2}\right)}{1 - \frac{\alpha}{8}}$$

$$\leq (69 - 37\alpha) + C\left(1 - \frac{\alpha}{4}\right).$$

Next we choose

$$\begin{cases} A & \geq (58 - 37\alpha) + A\left(1 - \frac{\alpha}{8}\right) + C \cdot \frac{3}{\alpha}; \\ C & \geq (69 - 37\alpha) + C\left(1 - \frac{\alpha}{4}\right); \end{cases}$$

$$\begin{cases} A & \geq \frac{464}{\alpha} + \frac{6624}{\alpha^3} \geq \frac{7088}{\alpha^3}; \\ C & \geq \frac{276}{\alpha}. \end{cases}$$

One can observe, that the coefficient in front of the $\left\|h^k\right\|^2$ is less, than zero:

$$4A(1 + s_1^{-1})(\gamma L)^2 - \frac{1 - \gamma\mu}{2} \leq \frac{1}{4} - \frac{1 - \gamma\mu}{2} = -\frac{1}{4} - \frac{\gamma\mu}{2} < 0,$$

with

$$\gamma \leq \min\left\{\frac{\alpha}{8\mu}, \frac{\alpha^2}{16\sqrt{1772}\sqrt{\frac{1}{M}\sum_{i=1}^{M} L_i^2}}\right\}.$$

Lastly, we get

$$\sum_{k=0}^{K-1} p^k \mathcal{Z}^{k+1} + \sum_{k=0}^{K-1} p^k A\gamma^2 \mathcal{F}^{k+1} + \sum_{k=0}^{K-1} p^k C\gamma^2 \mathbb{E}^{k+1} \leq \sum_{k=0}^{K-1} p^k(1 - \gamma\mu)\mathcal{Z}^k$$

$$+ \sum_{k=0}^{K-1} p^{k-1}\gamma^2 A\mathcal{F}^k$$

$$+ \sum_{k=0}^{K-1} p^{k-1}\gamma^2 C\mathbb{E}^k.$$

Using the choice of $p$ we derive the final estimate:

$$\sum_{k=0}^{K-1} (1 - \gamma\mu)^{-k}\mathcal{Z}^{k+1} + \sum_{k=0}^{K-1} (1 - \gamma\mu)^{-k}A\gamma^2\mathcal{F}^{k+1}$$

$$+ \sum_{k=0}^{K-1} (1 - \gamma\mu)^{-k}C\gamma^2\mathbb{E}^{k+1}$$

$$\leq \sum_{k=0}^{K-1}(1-\gamma\mu)^{-k+1}\mathcal{Z}^k$$

$$+ \sum_{k=0}^{K-1}(1-\gamma\mu)^{-k+1}\gamma^2 A\mathcal{F}^k$$

$$+ \sum_{k=0}^{K-1}(1-\gamma\mu)^{-k+1}\gamma^2 C\mathbb{E}^k;$$

$$(1-\gamma\mu)^{-K+1}\mathcal{Z}^K + (1-\gamma\mu)^{-K+1}A\gamma^2\mathcal{F}^K \quad + \quad (1-\gamma\mu)^{-K+1}C\gamma^2\mathbb{E}^K$$

$$\leq \quad (1-\gamma\mu)\mathcal{Z}^0$$

$$+ \quad (1-\gamma\mu)\gamma^2 A\mathcal{F}^0$$

$$+ \quad (1-\gamma\mu)\gamma^2 C\mathbb{E}^0;$$

$$V^K \quad \leq \quad (1-\gamma\mu)^K V^0,$$

where we denote $V^k = \mathcal{Z}^k + A\gamma^2\mathcal{F}^k + C\gamma^2\mathbb{E}^k$. Taking full expectation ends the proof of the theorem. $\square$

**Corollary 2.9** *Suppose the conditions of Theorem 2.8 hold. Then Algorithm 2 has*

$$\mathcal{O}\left(\left(\frac{\omega}{q_\alpha} + \frac{\hat{L}}{\mu\alpha^2 q_\alpha}\right)\log\left(\frac{1}{\varepsilon}\right)\right)$$

*communication complexity, where $q_\alpha$ is the expected density and $\varepsilon$ represents accuracy of the solution.*

*Proof.* Proof repeats the proof of Corollary 2.6. $\square$

## THE USE OF LARGE LANGUAGE MODELS (LLMs)

In this work, large language models (LLMs) were used exclusively for spelling edits.

