# OpenReview forum: "DIANA with Compression for Distributed Variational Inequalities: Eliminating the Need to Transmit Full Gradients"
_ICLR.cc/2026/Conference — Submitted to ICLR 2026_

### Official Review · Reviewer_KJMJ · 2025-10-29

**Soundness:** 3
**Presentation:** 3
**Contribution:** 3
**Rating:** 6
**Confidence:** 4

**Summary:**

The paper studies distributed variational inequalities (VIs) and proposes two extragradient–DIANA variants that avoid sending full gradients: (i) an unbiased-compression version (Algorithm 1, “EG-DIANA”) and (ii) a contractive-compression version with error feedback (Algorithm 2, “EG-DIANA-EF”). The methods combine DIANA’s control-variate mechanism with the extragradient step and, in the contractive case, add error compensation so that practical biased compressors (e.g., Top-$k$) can be used without requiring uncompressed “refresh” rounds. The analysis assumes Lipschitz continuity of local operators and strong monotonicity of the full operator; it establishes linear convergence and gives explicit communication complexities.

For EG-DIANA (Theorem 2.5 and Corollary 2.6), the method achieves
$$
\mathbb{E}\|z^{k}-z^\star\|^2 \le \big(1-\tfrac{\mu}{12L}\big)^k \|z^0-z^\star\|^2,
$$
and requires at most
$$
\mathcal{O}\Big[\frac{\omega}{q_\omega}+\frac{\omega L}{\mu q_\omega}+\frac{\omega^{3/2} \hat L}{\mu q_\omega\sqrt{M}}\Big]\log\frac{1}{\varepsilon}
$$
communications to reach accuracy $\varepsilon$.

For EG-DIANA-EF (Theorem 2.8 and Corollary 2.9) with contractive compressor parameter $\alpha$, the linear rate
$$
\mathbb{E}\|z^{k}-z^\star\|^2 \le \big(1-c\alpha\tfrac{\mu}{L}\big)^k \|z^0-z^\star\|^2
$$
leads to a total communication complexity of
$$
\mathcal{O}\Big[\frac{1}{q_\alpha \alpha}+\frac{\hat L}{\mu \alpha^2 q_\alpha}\Big] \log\frac{1}{\varepsilon}.
$$
In particular, for practical compressors such as Top-$k$ (where $q_\alpha\approx1/\alpha$), this becomes
$$
\mathcal{O}\Big(1+\frac{\hat L}{\mu \alpha}\Big)\log\frac{1}{\varepsilon}.
$$

Empirically, the paper reports CIFAR-10/ResNet-18 and GAN experiments showing that both methods converge faster per transmitted bit than MASHA2, EF21, and DIANA for compression levels between 1 % and 10 %, while MASHA requires occasional uncompressed transmissions that the proposed methods eliminate.

**Strengths:**

- **Problem relevance & positioning.** Eliminating periodic full-gradient transmissions in compressed methods for distributed VIs addresses a concrete limitation of prior art such as MASHA, which explicitly acknowledges the need for some uncompressed rounds
- **Technical contribution.** Clear integration of DIANA-type control variates with extragradient, and addition of error feedback to handle contractive compressors; assumptions and update rules are laid out and matched to the analysis.
- **Theory.** Linear convergence under strong monotonicity with explicit stepsize choices and Lyapunov potential; communication complexities are provided for both unbiased and contractive compressor regimes (Corollaries 2.6 and 2.9).
- **Empirical evidence.** CIFAR-10/ResNet-18 and GAN-style experiments show favorable time and data-transfer curves against DIANA/EF21/MASHA2, and the paper releases an anonymized code link.
- **Contextualization.** The manuscript situates the contribution relative to unbiased-compression DIANA, EF-style methods for biased compressors, and recent “EF-BV” unification efforts that recover DIANA and EF21 as special cases.


[1] Condat, Laurent, Kai Yi, and Peter Richtárik. "EF-BV: A unified theory of error feedback and variance reduction mechanisms for biased and unbiased compression in distributed optimization." Advances in Neural Information Processing Systems 35 (2022): 17501-17514.

**Weaknesses:**

- **Comparative baselines.** While MASHA2, DIANA, and EF21 are included, there is no quantitative comparison to EF-BV (NeurIPS’22), which explicitly unifies DIANA and EF21 and provides linear rates with a broader compressor class; such a baseline would sharpen the claimed advantages.
- **Assumption sharpness.** The main linear-rate results rely on strong monotonicity; it is unclear whether similar guarantees (e.g., last-iterate rates) extend to merely monotone VIs or Minty settings, which several VI works consider central. Clarify what breaks under weaker monotonicity and whether gap-function rates are available.
- **Complexity interpretation.** The communication bounds hide sizable constants and mixture terms (e.g., involving $\omega$, $q_\omega$, $\hat L$, $M$). A clearer discussion of dominating regimes and a comparison against state-of-the-art bidirectional-compression bounds (e.g., EF21-P) would help practitioners understand when the proposed methods are preferable.
- **Experimental realism.** The CIFAR-10 setup is a reformulated min-max proxy; details about non-IID partitions, partial participation, number of seeds, variance bars, and compressor wall-clock overheads are limited. Moreover, the statement that MASHA’s “full-gradient” steps double wall-clock time should be validated with matched operator-evaluation counts and identical hardware settings.
- **Ablations.** No ablations isolate the roles of (a) the extragradient step vs. single-gradient variants, (b) the control-variate $\beta$ parameter, and (c) the error-feedback buffer on stability with aggressive compression.


[2] Gruntkowska, Kaja, Alexander Tyurin, and Peter Richtárik. "EF21-P and friends: Improved theoretical communication complexity for distributed optimization with bidirectional compression." International conference on machine learning. PMLR, 2023.

**Questions:**

1. **On the necessity of error feedback:** For Algorithm 2, can the authors quantify when EF materially improves the contraction regime compared to a pure contractive-DIANA analysis (i.e., bounds in terms of $\alpha$, $q_\alpha$) and whether EF is still needed if one allows occasional refresh steps? Please relate to EF-BV’s bias–variance compressor class.
2. **Bidirectional compression:** Do the methods—and their proofs—extend to bidirectional compression with decoupled workers→server and server→workers noise (as in EF21-P), and what communication complexity would result in that setting?
3. **Weakly/merely monotone VIs:** Is there a proximal/extragradient variant that yields rates under **monotonicity** without strong monotonicity, perhaps via gap-function decay? If not, what obstacle in the proof prevents such an extension?
4. **Fairness of wall-clock comparisons:** In Fig. 1, are per-iteration costs normalized for the two operator evaluations of extragradient and for any compressor-specific sparsity/quantization kernels? Please report operator-call counts and compressor run-times alongside data-transfer MB.
5. **Relation to MASHA:** Beyond avoiding uncompressed rounds, how do the proposed rates compare to MASHA’s in matched assumptions (e.g., strong monotonicity), and do the authors foresee pathological instances where MASHA’s occasional refresh yields faster practical convergence?

**Details Of Ethics Concerns:**

No ethics concerns identified.

---

### Official Review · Reviewer_DATd · 2025-10-31

**Soundness:** 2
**Presentation:** 3
**Contribution:** 2
**Rating:** 4
**Confidence:** 3

**Summary:**

The paper proposes distributed extragradient methods based on DIANA to solve strongly monotone variational inequality (VI) problems under communication compression. The authors introduce two variants: extragradient DIANA with standard unbiased compressors and extragradient DIANA with error feedback for unbiased but contractive compressors. The paper establishes linear convergence guarantees for both variants under standard assumptions, including Lipschitz continuity and strong monotonicity of the operator, and the unbiasedness/contractiveness of the compression operators. Finally, the paper provides empirical evaluations, suggesting the efficiency of their proposed algorithms, on ResNet18 and GAN training, both of which can be formulated as minimax optimization problems corresponding to VI problems.

**Strengths:**

The paper proposes distributed extragradient methods based on DIANA, incorporating communication compression to solve strongly monotone variational inequality (VI) problems.  It establishes linear convergence guarantees for extragradient DIANA with any unbiased compressor, and for extragradient DIANA with error feedback under unbiased and contractive compressors.  The theoretical results are derived under standard assumptions, including Lipschitz continuity and strong monotonicity of the operator, as well as the unbiasedness and/or contractiveness of the compression operator. Finally, the empirical evaluation includes extensive experiments on ResNet18 and GAN training, both of which can be formulated as minimax optimization problems corresponding to VI problems.

**Weaknesses:**

**Lack of comparisons between DIANA SGDA [1] and the proposed algorithms:**

DIANA SGDA was analyzed by [1] and has the advantage of handling stochastic gradient cases. This is not the case for extragradient DIANA, thus representing a limitation of the proposed method.
- Relevant results from this literature should be included in Table 1, which currently omits them.
- Given the above reason and the fact that DIANA works well with any unbiased randomized compressors and in stochastic gradient settings, it is more valuable to extend the results of extragradient DIANA to handle stochastic gradients.
- Given that SDGA can potentially diverge for solving monotone variational inequality problems, whereas  extragradient methods can still converge, it is better to also analyze extragradient DIANA for monotone variational inequality problems.


**Bidirectional compression in MASHA vs. client-to-server compression in extragradient DIANA:**

MASHA 1 and MASHA 2 utilize bidirectional compression, whereas extragradient DIANA, with or without error feedback, only employs client-to-server compression. This limitation arises because extragradient DIANA requires the server to transmit \( z^{k+1/2} \) to all workers at each iteration. This restriction represents an additional limitation of the proposed approach.


**Why is error feedback needed for extragradient DIANA?**

It is unclear why error feedback is needed for extragradient DIANA (Algorithm 2), since Algorithm 1 already converges towards the true solution with any unbiased compressors.
- Error feedback is typically useful for biased but contractive compressors, but Theorems 2.5 and 2.8 do not clarify whether it improves convergence for extragradient DIANA.
- Additionally, no empirical comparison between extragradient DIANA with and without error feedback is provided. Including such results would help justify the necessity of error feedback in this context.

**Other weaknesses:**

- The empirical evaluations in Section 3.1 (ResNet18 training) and Section 3.3 (GAN training) likely involve problems whose fixed-point operators are not strongly monotone. It would strengthen the paper to include experiments on problems that satisfy the strongly monotone VI assumption.
- Since DIANA SGDA was proposed by [1], it would be valuable to include a comparison between this algorithm and extragradient DIANA.


[1] Beznosikov, A., Gorbunov, E., Berard, H., & Loizou, N. (2023, April). Stochastic gradient descent-ascent: Unified theory and new efficient methods. In _International conference on artificial intelligence and statistics_ (pp. 172-235). PMLR.

**Questions:**

1. Why use the proposed error feedback mechanism instead of other novel error feedback mechanisms like EF21 or EControl? Because EF21 and EControl are known to provide stronger convergence guarantees compared to the vanilla error feedback considered in this paper.

2. Rather than combining extragradient DIANA with error feedback, why not use an extragradient EF21 approach? Because EF21 can handle potentially biased but contractive compressors, which are more general than the unbiased and contractive compressors considered in Section 2.2.

---

### Official Review · Reviewer_YKrn · 2025-11-04

**Soundness:** 3
**Presentation:** 3
**Contribution:** 2
**Rating:** 2
**Confidence:** 4

**Summary:**

This work studied distributed VIs with compression operators. It proposed a new algorithm, which combines DIANA, EG and unbiased/contractive compressor. Compared to MASHA, the proposed algorithm does not need the full gradient computation for transmission. Convergence rate under smooth strongly-monotone cases are provided. Numerical experiments are complemented to verify the effectiveness of the proposed algorithm.

**Strengths:**

1. Extend MASHA and avoid full gradient computation and transmission.
2. Experiments reveal superior performances versus other algorithms.
3. The writing is easy to follow in general.

**Weaknesses:**

1. Current analysis only considered strongly-monotone cases, which largely restricted the applicability of the work, as a comparison, MASHA additionally considered both monotone and non-monotone/Minty cases. The extension from min to VIs is as expected, which also restricted the novelty of the work.
2. Also current algorithm works only in the deterministic case (each device compute its full gradient $F_i$), but stochastic algorithms is more popular now in ML applications, so I encourage authors to further the study into stochastic regime.
3. Experiments. You claimed to use the problem in Madry et al. (2017) as the problem. But if I understand it correctly, Madry et al. (2017) considered adversarial ML, which is NOT an "reformulation" as classical ERM problems. Also in that example, $\sigma$ comes with a compact domain in general, which is not displayed here. Currently the example is more like a modified model of Madry et al's work.
4. Bidirectional compression is missing. Although full gradient transmission is avoid in the proposed algorithm, but the server still needs to send uncompressed message to each client, which still can render the communication. Authors claimed to focus on gradient transmission, but the vector $z$ should share the same dimension number with the gradient, also there are already several bidirectional compression works. Maybe there are some difference between uplink and downlink, but it should be further clarified in the context.
5. Avoiding full gradient transmission is the main highlight in the contributions. But existing methods generally compute the full gradient periodically, with a very low frequency. It is better to add more results on the number of bits transmitted during iterations, to have a better understanding of the improvement on the transmission.

**Questions:**

See weaknesses.

---

### Official Review · Reviewer_MLxF · 2025-11-04

**Soundness:** 2
**Presentation:** 2
**Contribution:** 1
**Rating:** 2
**Confidence:** 5

**Summary:**

The authors have studied variational inequalities in distributed settings and combined DIANA with Error feedback mechanism. The theoretical results are established under strongly monotone operators and $L$-Lipschitz operators.

**Strengths:**

Studying variational inequalities in a distributed setting is a major and important problem to consider.

**Weaknesses:**

The main issue of this submission is the lack of comparison with highly relevant related work and baselines. All these works are highly relevant, focusing on unbiased compression, and variational inequalities:

[1] A.D. Nguyen, I. Markov, F. Z. Wu, A. Ramezani-Kebrya, K. Antonakopoulos, D. Alistarh, and V. Cevher. Layer-wise Quantization for Quantized Optimistic Dual Averaging. ICML 2025.

[2] I. Markov, K. Alim, E. Frantar, and D. Alistarh. L-GreCo: Layerwise-adaptive gradient compression for efficient
data-parallel deep learning. MLSys 2024.

[3] A. Beznosikov, E. Gorbunov, H. Berard, and N. Loizou. Stochastic gradient descent-ascent: Unified theory and
new efficient methods. AISTATS 2023.

[4] A. Ramezani-Kebrya, K. Antonakopoulos, I. Krawczuk, J. Deschenaux, and V. Cevher. Distributed extra-gradient with optimal complexity and communication guarantees. ICLR 2023.

[5] I. Markov, H. Ramezanikebrya, and D. Alistarh. CGX: adaptive system support for communication-efficient deep learning. In Proceedings of the 23rd ACM/IFIP International Middleware Conference 2022.

[6] A. Ramezani-Kebrya, F. Faghri, I. Markov, V. Aksenov, D. Alistarh, D. M. Roy. NUQSGD: Provably communication-efficient data-parallel SGD via nonuniform quantization. JMLR 2021.

[7] F. Faghri, I. Tabrizian, I. Markov, D. Alistarh, D. M. Roy, and A. Ramezani-Kebrya. Adaptive gradient quantization for data-parallel SGD. NeurIPS 2020.

[8] W. Wen, C. Xu, F. Yan, C. Wu, Y. Wang, Y. Chen, and H. Li. TernGrad: Ternary gradients to reduce communication in distributed deep learning. NeurIPS 2017.

**Questions:**

I would encourage the authors to fully rewrite the paper and properly cite and compare with relevant work.

---

### Meta-Review · Area_Chair_ixz3 · 2025-12-31

**Summary:**

This paper studied distributed variational inequalities based on compression operators, and proposed an extra-gradient DIANA (with error feedback) algorithm by combining DIANA, EG and unbiased/contractive compressor. In particular, the proposed algorithm does not need the full gradient computation for transmission. Meanwhile, this paper provided convergence analysis for the proposed method under smooth strongly-monotone setting. It also provided some numerical experiments to verify effectiveness of the proposed algorithm.

Overall, the proposed methods only simply combine some existing methods such as DIANA, EG and unbiased/contractive compressor,  so novelty of this paper is limited.  Meanwhile, many related works such as many minimax optimization algorithms do not be mentioned in the paper, and many comparisons such as DIANA SGDA are missing in the experiments.

The authors do not  address any reviewers'  concerns. Since all reviewers tend to reject this paper, I agree with this assessment.

**Reviewer Concerns:**

The authors do not provide any rebuttals to deal with the reviewers' concerns.

**Reviewer Scores:**

Reviewers MLxF,  YKrn, DATd suggest rejecting this paper. Since the authors do not address their concerns, they can not increase their scores. Although Reviewer  KJMJ suggest accepting it, his/her concerns also do not be addressed by the authors, so he/she also can not increase his/her score.  Overall, all reviewers tend to reject this paper. I agree with this assessment.

---

### Decision · Program_Chairs · 2026-01-26

Reject